# MaskTwins: Dual-form Complementary Masking for Domain-Adaptive Image Segmentation

**Jiawen Wang** [1 2] **Yinda Chen** [1 2 *] **Xiaoyu Liu** [1] **Che Liu** [3] **Dong Liu** [1] **Jianqing Gao** [4] **Zhiwei Xiong** [1 2]

## Abstract

Recent works have correlated Masked Image Modeling (MIM) with consistency regularization in Unsupervised Domain Adaptation (UDA). However, they merely treat masking as a special form of deformation on the input images and neglect the theoretical analysis, which leads to a superficial understanding of masked reconstruction and insufficient exploitation of its potential in enhancing feature extraction and representation learning. In this paper, we reframe masked reconstruction as a sparse signal reconstruction problem and theoretically prove that the dual form of complementary masks possesses superior capabilities in extracting domain-agnostic image features. Based on this compelling insight, we propose MaskTwins, a simple yet effective UDA framework that integrates masked reconstruction directly into the main training pipeline. MaskTwins uncovers intrinsic structural patterns that persist across disparate domains by enforcing consistency between predictions of images masked in complementary ways, enabling domain generalization in an end-to-end manner. Extensive experiments verify the superiority of MaskTwins over baseline methods in natural and biological image segmentation. These results demonstrate the significant advantages of MaskTwins in extracting domain-invariant features without the need for separate pre-training, offering a new paradigm for domain-adaptive segmentation. The source code is available at https://github.com/jwwang0421/masktwins.

---
[*]Theory contribution [1]University of Science and Technology of China [2]Institute of Artificial Intelligence, Hefei Comprehensive National Science Center [3]Data Science Institute, Imperial College London [4]iFLYTEK CO., LTD. Correspondence to: Zhiwei Xiong <zwxiong@ustc.edu.cn>, Jianqing Gao <jqgao@iflytek.com>.

*Proceedings of the $42^{st}$ International Conference on Machine Learning*, Vancouver, Canada. PMLR 267, 2025. Copyright 2025 by the author(s).

## 1. Introduction

Inspired by Masked Language Modeling (MLM) (Devlin, 2018; Brown, 2020) in natural language processing, Masked Image Modeling (MIM) (Bao et al., 2022; He et al., 2022; Xie et al., 2022b) has achieved remarkable success in self-supervised visual representation learning. MIM learns semantic representations by deliberately obscuring parts of the input and then reconstructing the missing information based on the unmasked parts, e.g., normalized pixels (He et al., 2022; Xie et al., 2022b), HOG feature (Wei et al., 2022), discrete tokens (Bao et al., 2022; Dong et al., 2023), deep features (Zhou et al., 2021; Dong et al., 2022) or frequencies (Xie et al., 2022a; Liu et al., 2023). The success of these methods can be attributed to their ability to force models to learn robust and generalizable features, even in the face of incomplete or corrupted input data. By simulating real-world scenarios where visual information may be partially occluded or distorted, masked reconstruction techniques enable models to develop a more comprehensive understanding of visual concepts.

Analogously, consistency regularization in unsupervised domain adaptive segmentation learns domain-invariant features by enforcing consistency between the predictions of transformed images and their original counterparts. In Unsupervised Domain Adaptation (UDA), consistency regularization based methods (Choi et al., 2019; Araslanov & Roth, 2021; Melas-Kyriazi & Manrai, 2021) typically utilize a variety of augmentations, like affine transformations, color jittering and cutout (DeVries, 2017). Recently, MIC (Hoyer et al., 2023) uses masked image consistency to learn context relations. Meanwhile, Shin et al. (2024) and Yang et al. (2025) have preliminarily explored the complementary masking in multi-modal tasks. However, these methods rely on specific multi-modal datasets and neglect the potential of complementary masking in a single modality. Moreover, the lacked theoretical analysis results in a cursory understanding of masked reconstruction and a failure to fully harness complementary contexts for feature extraction and representation learning.

In this paper, we propose a novel perspective on masked reconstruction by reframing it as a sparse signal reconstruction problem and utilize it to design an effective strategy

for domain-adaptive segmentation. Our theoretical analysis reveals that the dual form of complementary masks possesses superior image feature extraction capabilities. This insight is grounded in the principles of compressed sensing (Donoho, 2006), suggesting that complementary masks can provide a more comprehensive sampling of the input space. Building upon this theoretical foundation, we introduce MaskTwins, a simple yet effective framework for domain-adaptive segmentation. MaskTwins leverages the consistency constraints of complementary masks to extract domain-invariant features. This approach not only advances the theoretical understanding of masked reconstruction but also provides a practical framework for improving performance on domain-adaptive vision tasks.

Our contributions can be summarized as follows:

1. We provide a theoretical foundation for masked reconstruction by reframing it as a sparse signal reconstruction problem, offering new insights into the effectiveness of complementary masks. This perspective bridges the gap between masked image modeling and signal processing theory, potentially opening new avenues for future research.

2. We propose MaskTwins, a UDA framework that enforces consistency between predictions of dual-form complementary masked images without introducing extra learnable parameters. Furthermore, we study the complementary masking strategy and first prove its capability in extracting domain-invariant features. Our theoretical analysis provides a conceptual guidance for the broader application of complementary masking.

3. We demonstrate the superiority of our approach through extensive experiments, showing significant improvements over baseline methods in both natural and biological image segmentation. Our results indicate that MaskTwins can enhance model robustness and adaptability across diverse domains.

## 2. Related Works

**Unsupervised domain adaptation (UDA)** addresses the critical problem of performance degradation in target domains through the effective exploitation of both labeled source domain data and unlabeled target domain data. By bridging the domain gaps, UDA has emerged as a versatile solution to enhance model robustness in various computational domains, demonstrating promising results on various computer vision tasks such as natural image semantic segmentation (Tsai et al., 2018; Mei et al., 2020; Jiang et al., 2022) and medical image segmentation (Bermúdez-Chacón et al., 2018; Liu et al., 2020a; Wu et al., 2021). UDA solutions are broadly categorized into three groups: statistical

moment alignment (Chen et al., 2019; Liu et al., 2020b), adversarial learning (Tsai et al., 2018; Luo et al., 2021; Zheng & Yang, 2022) and self-training (Zou et al., 2018; Mei et al., 2020; Zhao et al., 2023). Methods based on statistical moment alignment aim to minimize the domain discrepancy employing an appropriate statistical distance function such as entropy minimization (Chen et al., 2019) and Wasserstein distance (Liu et al., 2020b). Adversarial training methods achieve domain invariant feature extraction with a GAN framework (Goodfellow et al., 2014). To overcome the challenges of instability in adversarial learning, Zheng & Yang (2022) adaptively refine the distribution of training data by aggregating the weak models. In self-training, pseudo labels (Lee et al., 2013) are created for the unlabeled target domain using confidence thresholds (Zou et al., 2018; 2019; Mei et al., 2020), pseudo-label prototypes (Zhang et al., 2019a; 2021; Jiang et al., 2022) or uncertainty (Zheng & Yang, 2021). Recently, Hoyer et al. (2023) explore context relations while Zhao et al. (2023) learn pixel-wise representations to boost the quality of pseudo-labels. Different from these UDA methods, our proposed MaskTwins integrates the context relationships by enforcing complementary masked consistency without introducing extra learnable parameters. The dual-form masked image consistency enables the learning of complementary clues, which further boosts the extraction of domain-invariant features.

**Masked Image modeling (MIM)** methods are showing great promise in visual self-supervised representation learning for their ability to learn robust and generalizable features from incomplete or corrupted input data, enhancing the models' comprehension of visual concepts. Many target signals have been conceived for the masked reconstruction, encompassing raw pixels (He et al., 2022; Xie et al., 2022b), HOG features (Wei et al., 2022), discrete visual tokens (Bao et al., 2022; Dong et al., 2023), frequencies (Xie et al., 2022a; Liu et al., 2023) and deep features (Zhou et al., 2021; Dong et al., 2022). Wang et al. (2023) further explore the reconstruction process at multiple scales while Kong & Zhang (2023) interpret MIM in a unified framework. However, these works mainly treat masked reconstruction as a pre-training strategy but neglect its potential for downstream tasks related to domain generalization. Hoyer et al. (2023) preliminarily explore the masked target image in the UDA setting and conclude that masked image consistency substantially boosts UDA performance through additional context clues. Recently, complementary masking has been superficially utilized with cross-modal thermal imaging (Shin et al., 2024) and depth estimation (Yang et al., 2025). Yet, a thorough theoretical foundation for the effectiveness of masked images in domain adaptation remains to be established. In this work, we introduce a novel reconceptualization of the masked reconstruction as a sparse signal reconstruction problem and refine the theory

of complementary masks. By surpassing the constraints of domain-specific customization, MaskTwins employs a strategic complementary masking technique on the input data, ensuring a more holistic and nuanced understanding of the intrinsical data patterns.

## 3. Theoretical Analysis

Consistency regularization typically leverages a rich set of augmentations. Nonetheless, focusing excessively on selecting the most appropriate parameters and perturbation functions makes them depart from the simple principle of consistency. Inspired by MIC (Hoyer et al., 2023), we expect the performance of masked consistency in UDA. We further take insights from the paradigm of masked reconstruction (Bao et al., 2022; He et al., 2022; Xie et al., 2022b) and present a theoretical analysis addressing the properties of masked training in visual tasks to provide a formal foundation for the complementary masking strategy. This analysis focuses on information preservation, generalization bounds, and feature consistency. Detailed proofs of all results are provided in Appendix A.

**Definition 1** (Complementary Mask). *Let $D \in \{0,1\}^{H \times W}$ be a binary matrix, where each element $D_{ij} \sim Bernoulli(0.5)$. The complementary mask pair is defined as $(D, 1-D)$, where $1$ is the all-ones matrix of size $H \times W$.*

**Definition 2** (Random Mask). *Let $R \in \{0,1\}^{H \times W}$ be a binary matrix where each element $R_{ij} \sim Bernoulli(0.5)$ independently. The random mask pair is defined as $(R_1, R_2)$, where $R_1$ and $R_2$ are independent random masks.*

**Assumption 1** (Visual Data Model). *The input image $X \in \mathbb{R}^{H \times W \times C}$ is generated by the model $X = S + E + N$, where $S$ represents a sparse signal component, $E$ represents environmental factors, and $N \sim \mathcal{N}(0, \sigma^2 I)$ is additive Gaussian noise.*

**Assumption 2** (Feature Extraction Framework). *We consider a feature extraction framework with the objective function:*

$$\mathcal{L}(f) = \mathbb{E}_X[\ell(f(X_1), f(X_2))], \qquad (1)$$

*where $f : \mathbb{R}^{H \times W \times C} \to \mathbb{R}^k$ is the feature extraction function, and $\ell : \mathbb{R}^k \times \mathbb{R}^k \to \mathbb{R}$ is the loss function.*

**Theorem 1** (Information Preservation). *For any input image $X$, define the information preservation metric $IP(X_1, X_2) = \frac{\langle f(X_1), f(X_2) \rangle}{\|f(X)\|^2}$. Then:*

$$(X_D, X_{1-D}) = (D \odot X, (1-D) \odot X) \qquad (2)$$

$$(X_{R_1}, X_{R_2}) = (R_1 \odot X, R_2 \odot X) \qquad (3)$$

$$\mathbb{E}[IP(X_D, X_{1-D})] \geq \mathbb{E}[IP(X_{R_1}, X_{R_2})] \qquad (4)$$

$$Var(IP(X_D, X_{1-D})) \leq Var(IP(X_{R_1}, X_{R_2})), \qquad (5)$$

*where $\odot$ denotes element-wise multiplication, (2) and (3) represent complementary masking images and random masking images respectively.*

**Theorem 2** (Generalization Bound). *Assume $\ell$ is $L$-Lipschitz and $f$ is $\beta$-smooth. For any $\delta \in (0,1)$, with probability at least $1 - \delta$:*

$$|\mathcal{L}(f) - \hat{\mathcal{L}}_n(f)|_{(D, 1-D)} \leq C_1 L \beta BA / \sqrt{n}, \qquad (6)$$

$$|\mathcal{L}(f) - \hat{\mathcal{L}}_n(f)|_{(R_1, R_2)} \leq C_2 L \beta B \left( A + \sqrt{HWC} \right) / \sqrt{n}, \qquad (7)$$

*where $A = 2 + \sqrt{\log(2/\delta)/2}$, $B = \sup_{X \in \mathcal{X}} \|X\|_F$, and $C_1$, $C_2$ are constants.*

**Theorem 3** (Feature Consistency). *Define the feature consistency error as $FCE(X_1, X_2) = \|f(X_1) - f(X_2)\|_2$. Then for any $\delta \in (0,1)$, with probability at least $1 - \delta$:*

$$FCE(X_D, X_{1-D}) \leq C_1 \sigma \sqrt{k \log(HWC/\delta)}, \qquad (8)$$

$$FCE(X_{R_1}, X_{R_2}) \leq C_2 \left( \sigma \sqrt{k \log(HWC/\delta)} + \|E\|_F \sqrt{k \log(HWC/\delta)/HWC} \right), \qquad (9)$$

*where $C_1$, $C_2$ are constants.*

**Remark 1.** *The theoretical results demonstrate the advantages of complementary masking. Specifically, complementary masks offer better information preservation, tighter generalization bounds, and improved feature consistency, compared to random masking. These properties are critical for extracting domain-invariant features, which are essential in cross-domain tasks such as domain adaptation.*

## 4. Method

### 4.1. Overview

The MaskTwins framework for UDA in semantic segmentation is detailed in Figure 1. The objective is to train a neural network $f_\theta$ that effectively generalizes to the target domain, given a labeled source domain dataset $X^S = \{(x_i^S, y_i^S)\}_{i=1}^{N_S} \subseteq \mathcal{D}^S$ and an unlabeled target domain dataset $X^T = \{x_j^T\}_{j=1}^{N_T} \subseteq \mathcal{D}^T$. The framework operates by generating two complementary masked versions of each target image $x_j^T$, denoted as $D \odot x_j^T$ and $(1-D) \odot x_j^T$, where $D$ is a binary mask. A teacher model $f_\phi$, updated via the Exponential Moving Average (EMA) of the student parameters, generates pseudo-labels for the target domain. The student's predictions, together with the pseudo-labels from the teacher model, are used to compute the target-domain losses, while a supervised loss is computed using the labeled source data. This iterative process adapts the model to the target domain, leveraging both the supervised source information and the unsupervised adaptation to the target domain.

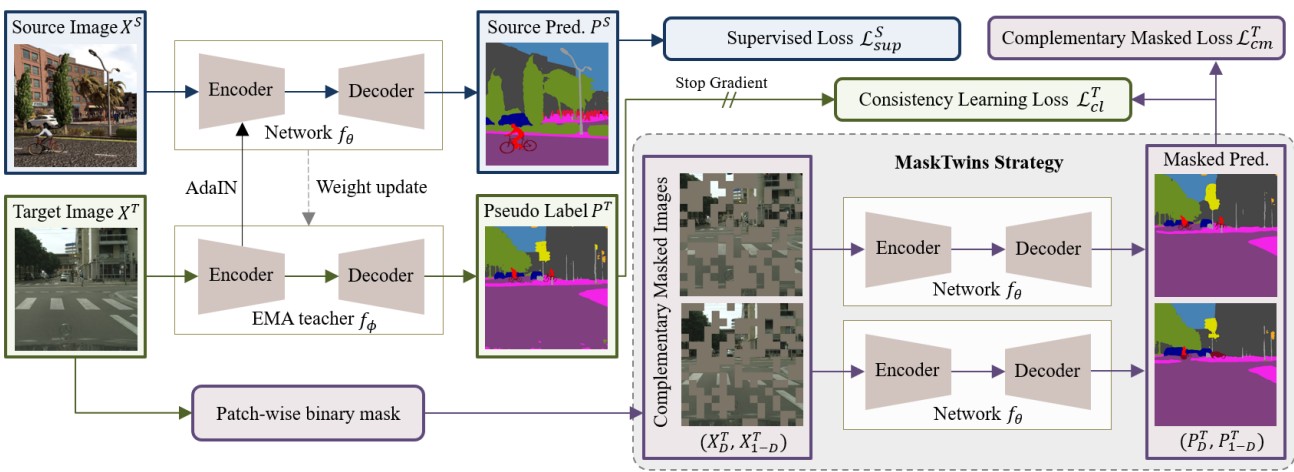

Figure 1: The overall framework of MaskTwins. Given the labeled source data, we calculate the segmentation prediction $P^S$ with the network $f_\theta$, supervised by basic segmentation loss $\mathcal{L}_{sup}^S$. For the target domain, we obtain the predictions of complementary masked target images, constrained by the pseudo-labels $P_T$ that are generated based on the unmasked image by an exponential moving average (EMA) teacher $f_\phi$. Furthermore, MaskTwins proposes the complementary masked loss between dual-form complementary masked images for deep consistency learning.

## 4.2. Complementary Masked Learning

Building upon the theoretical analysis, we now describe the core complementary masked learning approach in MaskTwins. This strategy employs patch-wise binary masks to generate dual complementary views of the target images. Specifically, for each target image $x_j^T$, a binary mask $D$ is sampled from a Bernoulli distribution:

$$D_{\substack{mb+1:(m+1)b \\ nb+1:(n+1)b}} \sim \text{Bernoulli}(1-r), \qquad (10)$$

where $r$ is the mask ratio, $b$ is the patch size, and $m$ and $n$ are patch indices. The dual-form complementary masked images are then obtained by element-wise multiplication:

$$X_{cm}^T = \{X_D^T, X_{1-D}^T\} = \{D \odot X^T, (1-D) \odot X^T\}. \quad (11)$$

These complementary views encourage the model to extract robust, domain-invariant features by enforcing consistency learning upon masked images. To effectively learn from dual-form complementary contexts, we introduce two kinds of consistency losses. First, we constrain the consistent prediction of complementary masked images, which enables the network to integrate the dual-form clues. The complementary masked loss is accordingly defined as:

$$\mathcal{L}_{cm}^T = \mathbb{E}[\|p_{j,D}^T, p_{j,1-D}^T\|^2], \qquad (12)$$

where $p_{j,D}^T$ and $p_{j,1-D}^T$ are the predictions for the complementary masked images. Intended to encourage successful masked reconstruction for both masked views, we also define a masked consistency learning loss:

$$\mathcal{L}_{cl}^T = \mathbb{E}[\lambda \times \mathcal{L}_{ce}(p_{j,D}^T, \hat{y}_j^T) + (1-\lambda) \times \mathcal{L}_{ce}(p_{j,1-D}^T, \hat{y}_j^T)], \qquad (13)$$

where $\hat{y}_j^T$ are the pseudo-labels, $\mathcal{L}_{ce}(\cdot, \cdot)$ refers to the standard cross entropy loss, $\lambda$ defaults to 0.5 to ensure balanced learning from the complementary masks. Since there is no ground truth available for the target domain, a teacher model $f_\phi$ predicts the pseudo-label for the unmasked target image:

$$\hat{y}_j^T = [c = \text{argmax} f_\phi(x_j^T)], \qquad (14)$$

where $c$ is one category and the pseudo-label is converted into a one-hot categorical form via the Iverson bracket $[\cdot]$.

The parameters of the teacher network $f_\phi$ are updated using an Exponential Moving Average (EMA) of the parameters of the student network $f_\theta$ (Tarvainen & Valpola, 2017):

$$\phi_{t+1} \leftarrow \alpha \phi_t + (1-\alpha)\theta_t, \qquad (15)$$

where $t$ denotes a training step and $\alpha$ is the EMA decay rate. The teacher model averages the weights of previous student models over time, leading to temporally stable and reliable predictions on the target domain.

This complementary masking strategy ensures that the model learns from diverse, yet consistent, views of the target domain, promoting robust generalization to the target domain. The next section details the overall model architecture and training process, which integrates these complementary masking principles.

## 4.3. Model Architecture and Training Strategy

The MaskTwins architecture consists of a shared encoder and segmentation head for both the source and target domains. To mitigate domain shift, we employ an Adaptive Instance Normalization (AdaIN) (Huang & Belongie, 2017)

module in the shallow layers of the network, which aligns feature distributions between the two domains.

During training, we apply the complementary masks to the target domain images and enforce consistency between the predictions of these masked versions. This encourages the model to learn invariant representations that generalize well to the target domain. Our training strategy integrates supervised learning on the source domain with self-training and consistency regularization on the target domain.

The supervised loss on the source domain is defined as:

$$\mathcal{L}_{sup}^S = \mathbb{E}[\mathcal{L}_{ce}(p_i^S, y_i^S)] = \mathbb{E}[-y_i^S \log(p_i^S)], \qquad (16)$$

where $p_i^S = f_\theta(x_i^S)$ is the source prediction of the network $f_\theta$ and $y_i^S$ is the corresponding ground truth.

By integrating these components - complementary masking, consistency regularization, and self-training with a teacher model - MaskTwins effectively leverages the complementary information from masked inputs, promoting robust feature learning and improved generalization.

The overall loss function that encapsulates our training strategy is formulated as:

$$\mathcal{L}_{total} = \mathcal{L}_{sup}^S + \mathcal{L}_{cl}^T + \lambda_{cm}\mathcal{L}_{cm}^T, \qquad (17)$$

where $\mathcal{L}_{sup}^S$ is the supervised loss on the source domain, $\mathcal{L}_{cl}^T$ is the masked consistency learning loss on the target domain, $\mathcal{L}_{cm}^T$ is the complementary masked loss, and $\lambda_{cm}$ is the weight for the complementary masked loss. We summarize the pipeline of MaskTwins in Algorithm 1 in Appendix E.

## 5. Experiments

### 5.1. Implementation details

**Datasets.** To demonstrate the versatility of MaskTwins, we conduct experiments spanning six distinct datasets: SYN-THIA (Ros et al., 2016) and Cityscapes (Cordts et al., 2016) are natural datasets, VNC III (Gerhard et al., 2013), Lucchi (Lucchi et al., 2013), MitoEM (Wei et al., 2020) and WASP-SYN (Li et al., 2024) are biological datasets. The details of the datasets and the task-specific implementation on these datasets can be found in Appendix F.

**MaskTwins parameters.** MaskTwins uses the square mask for 2D domain adaptation and the cube mask for 3D respectively. The complementary masks have equal loss weight and the same mask ratio $r = 0.5$. The mask patch size is fixed for each task, approximately $1/16$ of the input size. For SYNTHIA→Cityscapes, we use a patch size $b = 64$, a loss weight $\lambda_{cm} = 0.01$, and common color augmentation following the parameters of DAFormer (Hoyer et al., 2022a) and HRDA (Hoyer et al., 2022b). For mitochondria semantic segmentation, we use a patch size $b = 32$,

a loss weight $\lambda_{cm} = 0.01$, a pseudo-label threshold $\delta = 0.7$, and random augmentation including flip, transpose, rotate, resize and elastic transformation. For synapse detection, the point annotations (3D coordinates) are transformed into voxel cubes with a size of $3 \times 3 \times 3$ to be used as the training target. We use a patch size $b = 6$, a loss weight $\lambda_{cm} = 0.1$ Empirically, we set the threshold $\delta_{pre} = 0.75$ for the pre-synapse, $\delta_{post} = 0.65$ for the post-synapse by default. The experiments are conducted on $8\times$ RTX 3090 GPU.

### 5.2. Natural Image Semantic Segmentation

First, we compare MaskTwins with previous UDA methods on SYNTHIA→Cityscapes in Table 1. It can be seen that MaskTwins outperforms the previously state-of-the-art method by a significant margin of +2.7 mIoU and remains competitive in segmenting almost all classes, which verifies the effectiveness of the dual form of complementary masks on target images. Classes that most profit from our method are *sidewalk*, *road*, *vegetation*, *bus*, and *rider*. Particularly, *sidewalk* owns the lowest UDA performance over 13 categories, meaning that it is the most difficult to adapt for previous methods. Here, contextual relationships seem to be crucial for achieving successful adaptation. However, we increase the IoU of the *sidewalk* by +19.6 from 50.5 to 70.1 IoU. Additionally, our performance improvement on *road* is +4.8 from 91.2 to 96.0 IoU, probably because of its strong correlation with *sidewalk*. For some classes, our method increases the performance by a smaller margin or causes a minor reduction, probably because the small objectives lead MaskTwins to misunderstand the complementary masked regions. In Figure 2, we visualize the segmentation results and the comparison with previous strong methods HRDA (Hoyer et al., 2022b), MIC (Hoyer et al., 2023) and the ground truth. While previous methods are confused by illumination as well as crossings and fail to distinguish *sidewalk* from *road*, MaskTwins enables a more robust recognition of these categories. We can conclude that the complementary masking significantly enhances semantic segmentation, particularly for large or complex objects, where it effectively preserves structure and enables accurate segmentation despite obstacles, enhancing visual comprehension.

### 5.3. Mitochondria Semantic Segmentation

We conduct quantitative comparison results of our approach with multiple UDA baselines on the Lucchi and MitoEM datasets to demonstrate the superiority of our approach. As listed in Table 2, MaskTwins achieves the new state-of-the-art results in all cases, which corroborates the effectiveness of the proposed complementary masking strategy. Specifically, MaskTwins enhances the IoU of VNC III→Lucchi(Subset1) and Lucchi(Subset2) to 75.0% and 78.6%, outperforming the state-of-the-art methods by 3.2% and 3.2%. On the MitoEM dataset with a larger structure

Table 1: Comparison results with previous UDA methods on SYNTHIA→Cityscapes. "SW" stands for *sidewalk*, "TL" for *traffic light*, "TS" for *traffic sign*, "Veg." for *vegetation*, "PR" for *person*. We present per-class IoU and mean IoU (mIoU), averaged across 13 categories. The competitors include DAFormer (Hoyer et al., 2022a), CAMix (Zhou et al., 2022b), HRDA (Hoyer et al., 2022b), MIC (Hoyer et al., 2023), etc. More details are shown in Appendix D.

| Method | Road | SW | Build | TL | TS | Veg. | Sky | PR | Rider | Car | Bus | Motor | Bike | mIoU |
|---|---|---|---|---|---|---|---|---|---|---|---|---|---|---|
| SIBAN | 82.5 | 24.0 | 79.4 | 16.5 | 12.7 | 79.2 | 82.8 | 58.3 | 18.0 | 79.3 | 25.3 | 17.6 | 25.9 | 46.3 |
| DADA | 89.2 | 44.8 | 81.4 | 8.6 | 11.1 | 81.8 | 84.0 | 54.7 | 19.3 | 79.7 | 40.7 | 14.0 | 38.8 | 49.8 |
| BDL | 86.0 | 46.7 | 80.3 | 14.1 | 11.6 | 79.2 | 81.3 | 54.1 | 27.9 | 73.7 | 42.2 | 25.7 | 45.3 | 51.4 |
| APODA | 86.4 | 41.3 | 79.3 | 22.6 | 17.3 | 80.3 | 81.6 | 56.9 | 21.0 | 84.1 | 49.1 | 24.6 | 45.7 | 53.1 |
| FDA | 79.3 | 35.0 | 73.2 | 19.9 | 24.0 | 61.7 | 82.6 | 61.4 | 31.1 | 83.9 | 40.8 | 38.4 | 51.1 | 52.5 |
| CCM | 79.6 | 36.4 | 80.6 | 22.4 | 14.9 | 81.8 | 77.4 | 56.8 | 25.9 | 80.7 | 45.3 | 29.9 | 52.0 | 52.9 |
| LDR | 85.1 | 44.5 | 81.0 | 16.4 | 15.2 | 80.1 | 84.8 | 59.4 | 31.9 | 73.2 | 41.0 | 32.6 | 44.7 | 53.1 |
| CD-SAM | 82.5 | 42.2 | 81.3 | 18.3 | 15.9 | 80.6 | 83.5 | 61.4 | 33.2 | 72.9 | 39.3 | 26.6 | 43.9 | 52.4 |
| ASA | 91.2 | 48.5 | 80.4 | 5.5 | 5.2 | 79.5 | 83.6 | 56.4 | 21.9 | 80.3 | 36.2 | 20.0 | 32.9 | 49.3 |
| DAST | 87.1 | 44.5 | 82.3 | 13.9 | 13.1 | 81.6 | 86.0 | 60.3 | 25.1 | 83.1 | 40.1 | 24.4 | 40.5 | 52.5 |
| UncerDA | 79.4 | 34.6 | 83.5 | 32.1 | 26.9 | 78.8 | 79.6 | 66.6 | 30.3 | 86.1 | 36.6 | 19.5 | 56.9 | 54.6 |
| RPLR | 81.5 | 36.7 | 78.6 | 20.7 | 23.6 | 79.1 | 83.4 | 57.6 | 30.4 | 78.5 | 38.3 | 24.7 | 48.4 | 52.4 |
| UACR | 85.5 | 42.5 | 83.0 | 20.9 | 25.5 | 82.5 | 88.0 | 63.2 | 31.8 | 86.5 | 41.2 | 25.9 | 50.7 | 55.9 |
| DACS | 80.6 | 25.1 | 81.9 | 22.7 | 24.0 | 83.7 | 90.8 | 67.6 | 38.3 | 82.9 | 38.9 | 28.5 | 47.6 | 54.8 |
| ProDA | 87.8 | 45.7 | 84.6 | 54.6 | 37.0 | 88.1 | 84.4 | 74.2 | 24.3 | 88.2 | 51.1 | 40.5 | 45.6 | 62.0 |
| DAFormer | 84.5 | 40.7 | 88.4 | 55.0 | 54.6 | 86.0 | 89.8 | 73.2 | 48.2 | 87.2 | 53.2 | 53.9 | 61.7 | 67.4 |
| CAMix | 87.4 | 47.5 | 88.8 | 55.2 | 55.4 | 87.0 | 91.7 | 72.0 | 49.3 | 86.9 | 57.0 | 57.5 | 63.6 | 69.2 |
| HRDA | 85.2 | 47.7 | 88.8 | 65.7 | 60.9 | 85.3 | 92.9 | 79.4 | 52.8 | 89.0 | 64.7 | 63.9 | **64.9** | 72.4 |
| MIC | 86.6 | 50.5 | 89.3 | 66.7 | **63.4** | 87.1 | **94.6** | 81.0 | 58.9 | 90.1 | 61.9 | 67.1 | 64.3 | 74.0 |
| Ours | **96.0** | **70.1** | **89.5** | 66.8 | 62.1 | **89.1** | 94.3 | **81.5** | 59.7 | **90.5** | **66.6** | **67.7** | 63.6 | **76.7** |

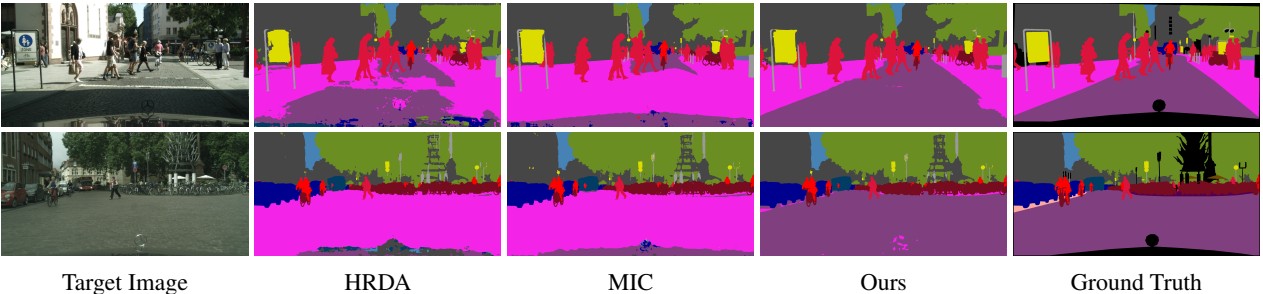

Figure 2: Qualitative segmentation results on SYNTHIA→Cityscapes. MaskTwins significantly improves the segmentation results of classes such as *sidewalk*, *road*, *bus* and *rider*.

discrepancy, our method consistently has remarkable improvements by +2.1% IoU and +1.3% IoU respectively. It is noticeable that the mitochondria in MitoEM-H exhibit denser and more intricate distributions compared to those in MitoEM-R, rendering the domain adaptation from MitoEM-R to MitoEM-H more challenging than the reverse. Despite this, MaskTwins surpasses CAFA (Yin et al., 2023) by a significant margin on the benchmark of MitoEM-R→MitoEM-H. It demonstrates that the proposed strategy can strengthen the generalization capacity of the learned model and adapt it to the challenging and diverse target domain. In Figure 3, we

further qualitatively compare MaskTwins with other competitive methods including DAMT-Net (Peng et al., 2020), DA-VSN (Guan et al., 2021), DA-ISC (Huang et al., 2022b), and CAFA (Yin et al., 2023). The results highlighted by yellow boxes reveal that MaskTwins shows better adaptability while other methods fail to handle hard cases with large domain gap. By leveraging the complementary masked context, our method manages to separate mitochondria correctly from the background and delivers more fine-grained results on the target domain. This indicates that MaskTwins is adept at extracting robust features of segmented objec-

Table 2: Quantitative comparisons on VNC III→Lucchi-Subset1 (V2L1), VNC III→Lucchi-Subset2 (V2L2), MitoEM-R→MitoEM-H (R2H) and MitoEM-H→MitoEM-R (H2R), metrics in %. "Oracle" denotes the model is trained on target with groundtruth labels, while "NoAdapt" represents the model pretrained on source is directly applied in target for inference without any adaptation strategy. The results of Oracle, NoAdapt, UALR, DAMT-Net, DA-VSN and DA-ISC are adopted from Huang et al. (2022b).

| Methods | V2L1 | | | | V2L2 | | | | R2H | | | | H2R | | | |
|---|---|---|---|---|---|---|---|---|---|---|---|---|---|---|---|---|
| | IoU | F1 | MCC | mAP | IoU | F1 | MCC | mAP | IoU | F1 | MCC | mAP | IoU | F1 | MCC | mAP |
| Oracle | 86.5 | 92.7 | 86.5 | - | 88.6 | 93.9 | - | - | 84.5 | 91.6 | 91.2 | 97.0 | 87.3 | 93.2 | 92.9 | 98.2 |
| NoAdapt | 40.3 | 57.3 | 40.3 | - | 44.3 | 61.3 | - | - | 39.6 | 56.8 | 59.2 | 74.6 | 61.9 | 76.5 | 76.8 | 88.5 |
| Advent | 59.7 | 74.8 | 73.3 | 78.9 | 70.7 | 82.8 | 81.8 | 90.5 | 69.6 | 82.0 | 81.3 | 89.7 | 74.6 | 85.4 | 84.8 | 93.5 |
| UALR | 57.0 | 72.5 | 71.2 | 80.2 | 65.2 | 78.8 | 77.7 | 87.2 | 72.2 | 83.8 | 83.2 | 90.7 | 75.9 | 86.3 | 85.5 | 92.6 |
| DAMT-Net | 60.0 | 74.7 | 60.0 | - | 68.7 | 81.3 | - | - | 73.0 | 84.4 | 83.7 | 92.1 | 75.4 | 86.0 | 85.7 | 94.8 |
| DA-VSN | 60.3 | 75.2 | 73.9 | 82.8 | 71.1 | 83.1 | 82.2 | 91.3 | 71.4 | 83.3 | 82.6 | 91.6 | 76.5 | 86.7 | 86.3 | 94.5 |
| DA-ISC | 68.7 | 81.3 | 80.5 | 89.5 | 74.3 | 85.2 | 84.5 | 92.4 | 74.8 | 85.6 | 84.9 | 92.6 | 79.4 | 88.5 | 88.3 | 96.8 |
| CAFA | 71.8 | 83.4 | 82.8 | 91.1 | 75.4 | 85.8 | 85.4 | 94.8 | 76.3 | 86.6 | 86.0 | 92.8 | 80.6 | 89.2 | 88.9 | 96.8 |
| Ours | **75.0** | **85.6** | **85.1** | **92.4** | **78.6** | **87.9** | **87.4** | **95.2** | **78.4** | **87.9** | **87.4** | **94.0** | **81.9** | **90.0** | **89.7** | **96.9** |

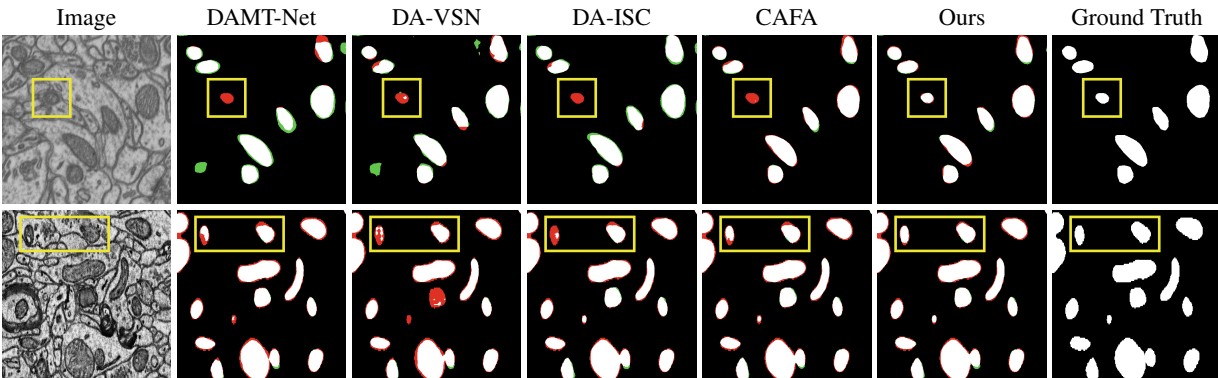

Figure 3: Qualitative comparison of MaskTwins with previous methods on VNC III→Lucchi Subset2 (row 1) and MitoEM-H→MitoEM-R (row 2). The pixels in red and green denote the false-negative and false-positive results respectively.

tives, thereby achieving effective adaptation from the source domain to the target domain.

### 5.4. Synapse Detection

We also evaluate the effectiveness of our proposed method on 3D synapse detection. This task aims to pinpoint the positions of pre-synaptic and post-synaptic sites in the 3D space, as well as to determine the connectivity between them, specifically identifying the IDs of the pre-synapses to which the post-synapses are linked. To further understand the complexity, it's worth noting that synapses often appear in highly cluttered regions, making accurate identification even more challenging. For a more vivid depiction of the detection outcomes, we visualize the 3D results of pre- and post-synapse detection in Appendix G.2. Following Chen et al. (2024), we convert the task of synapse detection into a segmentation task. Since there is few prior works on this

new challenge, we re-implement SSNS-Net (Huang et al., 2022a), AdaSyn (Chen et al., 2024) and MIC (Hoyer et al., 2023) strictly following their experimental implementations and make a fair comparison. Table 3 shows that MaskTwins achieves the highest F1-score with an outstanding gain of 2.02% totally, 3.13% on post-synapse. Due to its high density and the one-to-many synapse connectivity problem, post-synapses are more difficult to identify. Other methods perform poorly on post-synapse detection with a sharp decline in the performance of detecting the post-synapses. However, MaskTwins can learn robust features and capture more post-synapses correctly with the help of complementary masks. The results demonstrate that the dual-form complementary masking is effective even in the task of 3D synapse detection, showing its remarkable capability in extracting domain-invariant features across various domains and increases the robustness of network.

Table 3: Comparison result on the WASPSYN Challenge. The F1-score is the average of $F1_{pre}$ and $F1_{post}$. The competitors are SSNS-Net (Huang et al., 2022a), AdaSyn (Chen et al., 2024) and MIC (Hoyer et al., 2023).

| Method | $F1_{pre}$ | $F1_{post}$ | F1-score |
|---|---|---|---|
| SSNS-Net | 0.7201 | 0.3072 | 0.5137 |
| AdaSyn | 0.7846 | 0.3136 | 0.5491 |
| MIC | 0.7823 | 0.3599 | 0.5711 |
| MaskTwins(Ours) | **0.7914** | **0.3912** | **0.5913** |

Table 4: Ablation study of the components and masking strategy of MaskTwins on SYNTHIA→Cityscapes, including target-domain consistency learning loss (CL), complementary masking strategy (CMask), random masking strategy (RMask) as an alternative, EMA teacher and AdaIN.

| CL | CMask | RMask | EMA | AdaIN | mIoU |
|---|---|---|---|---|---|
| - | - | - | - | - | 53.7 |
| ✓ | - | - | ✓ | ✓ | 72.8 |
| ✓ | - | ✓ | - | - | 74.3 |
| ✓ | - | ✓ | - | ✓ | 74.6 |
| ✓ | - | ✓ | ✓ | - | 75.0 |
| ✓ | - | ✓ | ✓ | ✓ | 75.2 |
| ✓ | ✓ | - | - | - | 76.0 |
| ✓ | ✓ | - | - | ✓ | 76.1 |
| ✓ | ✓ | - | ✓ | - | 76.4 |
| ✓ | ✓ | - | ✓ | ✓ | **76.7** |

## 5.5. Ablation Study

**Component Ablation.** First, we ablate each component of MaskTwins on SYNTHIA→Cityscapes in Table 4. When the target-domain information is lacked, the performance is only 53.7 mIoU, since the source model trained with a single supervised loss has a poor ability of generalization. Compared to MIC, random masking yields a improvement of +1.2 mIoU, which demonstrates the effectiveness of the dual-form masking consistency. Replacing the random masking strategy with the proposed complementary masking strategy, the complete MaskTwins achieves 76.7 mIoU, which is +3.9 mIoU better than the domain adaptive baseline with consistency learning loss. On the other side, without the EMA teacher and AdaIN module, the performance only reduces by -0.6 mIoU and -0.3 mIoU, respectively. Therefore, masking strategy contributes most to the improvements among other used components of MaskTwins. And the complementary masking can achieve significantly higher improvement by only changing the masking strategy, further validating the substantial performance boost brought about by the complementary masking and the total gain does not rely on the masking strategy itself. The experiment results unequivo-

Table 5: Ablation study of the patch size $b$ and the mask ratio $r$ of MaskTwins on SYNTHIA→Cityscapes with a input size of 1024. Specially, the mask ratio $r$ indicates a complementary combination of $[r, 1-r]$.

(a) Mask Ratio.

| Mask Ratio | mIoU |
|---|---|
| 0.1 | 72.0 |
| 0.2 | 74.6 |
| 0.3 | 75.4 |
| 0.4 | 76.5 |
| 0.5 | **76.7** |

(b) Patch Size.

| Patch Size | mIoU |
|---|---|
| 32 | 76.2 |
| 64 | **76.7** |
| 128 | 75.9 |
| 256 | 75.6 |
| 512 | 75.0 |

cally substantiate the proved theorems in Section 3, showing the effectiveness of complementary masking strategy compared to random one.

**Patch Size and Mask Ratio.** Table 5 demonstrates the effect of the mask patch size $b$ and mask ratio $r$ on SYNTHIA→Cityscapes with a input size of 1024. We systematically alter the mask ratios and specifically explore the combinations of $[r, 1-r]$ with a mask ratio $r \in \{0.1, 0.2, 0.3, 0.4, 0.5\}$ in a complementary form. By gradually increasing the mask patch size, we observe that the best performance is achieved when $b = 64$, i.e., 1/16 of the input size. Patches that are either larger or smaller exhibit varying degrees of performance reduction. This is likely because patches that are too large may excessively cover the foreground while those that are too small tend to apply an overly dense masking, potentially hindering the complementary learning of contextual information. On the contrary, by concentrating on context-rich areas using appropriate mask patch size, the model can better utilize the spatial relations within the image, leading to improved performance in unsupervised domain adaptation. For mask ratio, we observe that the performance gets better when increasing the mask ratio $r$ to 0.5. Compared to MIC, MaskTwins consistently achieves significant improvements in a range of $b$ between 32 and 256 and $r$ between 0.2 and 0.5. Only for a mask ratio $r$ of 0.1, MaskTwins decreases the performance. The best performance is achieved when $r = 0.5$ and $b = 64$, i.e., 1/16 of the input size. Therefore, we use these parameter settings in all experiments.

## 6. Conclusion

In this work, we present a novel perspective on masked reconstruction by reinterpreting it as a sparse signal reconstruction problem and theoretically prove the effectiveness of the dual form of complementary masks. Based on this theoretical foundation, we propose MaskTwins, an effective framework that utilizes complementary masks to simulta-

neously enhance the robust feature extraction for domain-adaptive segmentation. Our MaskTwins has demonstrated remarkable superiority over the state-of-the-art methods across a diverse range of domain adaptation scenarios, spanning from natural to biological imaging and from 2D to 3D modalities. For instance, MaskTwins respectively achieves significant performance improvements by +2.7% and +2.5% in IoU on SYNTHIA→Cityscapes and biological datasets. Since MaskTwins performs masked image consistency without extra annotations, it offers a flexible technique that can be seamlessly incorporated with other methods to further facilitate the learning of domain-invariant features, ensuring the cross-domain knowledge adaptation process. In the future, we will continue to explore the potential of MaskTwins in a broader spectrum of visual recognition challenges, including but not limited to domain-adaptive video segmentation and image classification.

## Acknowledgements

This work was supported in part by the National Natural Science Foundation of China under Grants 624B2137 and 62021001.

## Impact Statement

This work aims to advance the field of machine learning. While the research may have various societal implications, we deem it unnecessary to highlight specific ethical or social impacts herein.

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

# A. Theory Proofs

## A.1. Complementary Masking Theory: Mean and Variance Analysis

**Definition 1: Complementary Mask**   Let $D \in \{0,1\}^d$ be a random binary vector where each element $D_i$ is independently drawn from Bernoulli$(0.5)$. The **complementary mask** is $1 - D$, where $1$ is the vector of ones in $\mathbb{R}^d$.

**Definition 2: Random Masks**   Let $D_1, D_2 \in \{0,1\}^d$ be independent random binary vectors where each element $D_{ki}$ (for $k = 1, 2$) is independently drawn from Bernoulli$(0.5)$. These are the **random masks**.

## A.2. Information Preservation Metric

Given a deterministic vector $x \in \mathbb{R}^d$, we define masked versions of $x$ as:

- For complementary masks:

$$x_1 = D \odot x, \quad x_2 = (1 - D) \odot x$$

- For random masks:

$$x_1 = D_1 \odot x, \quad x_2 = D_2 \odot x$$

where $\odot$ denotes element-wise (Hadamard) product.

Define the **information preservation (IP) metric** as:

$$\mathrm{IP}(x_1, x_2) = \frac{\langle x_1, x_2 \rangle}{\|x\|^2}$$

## A.3.  Mean and Variance Computations

### A.3.1. COMPLEMENTARY MASKS

**Mean:**

For complementary masks, note that for each coordinate $i$:

$$D_i(1 - D_i) = 0$$

because $D_i$ is either 0 or 1.

Therefore, the inner product:

$$\langle x_1, x_2 \rangle = \sum_{i=1}^{d} D_i(1 - D_i)x_i^2 = 0$$

Thus,

$$\mathrm{IP}(x_1, x_2) = \frac{0}{\|x\|^2} = 0$$

and

$$\mathbb{E}[\mathrm{IP}(x_1, x_2)] = 0$$

**Variance:**

Since $\mathrm{IP}(x_1, x_2) = 0$ almost surely,

$$\mathrm{Var}(\mathrm{IP}(x_1, x_2)) = 0$$

### A.3.2. RANDOM MASKS

**Mean:**

For random masks:

$$\langle x_1, x_2 \rangle = \sum_{i=1}^{d} D_{1i} D_{2i} x_i^2$$

Since $D_{1i}, D_{2i}$ are independent Bernoulli(0.5), we have:

$$\mathbb{E}[D_{1i} D_{2i}] = \left(\frac{1}{2}\right) \left(\frac{1}{2}\right) = \frac{1}{4}$$

Therefore,

$$\mathbb{E}[\langle x_1, x_2 \rangle] = \frac{1}{4} \|x\|^2$$

and

$$\mathbb{E}[\text{IP}(x_1, x_2)] = \frac{1}{4}$$

**Variance:**

Compute $\text{Var}(D_{1i} D_{2i})$:

$$\text{Var}(D_{1i} D_{2i}) = \frac{1}{4} - \left(\frac{1}{4}\right)^2 = \frac{3}{16}$$

Then,

$$\text{Var}(\langle x_1, x_2 \rangle) = \sum_{i=1}^{d} \frac{3}{16} x_i^4 = \frac{3}{16} \sum_{i=1}^{d} x_i^4$$

Thus,

$$\text{Var}(\text{IP}(x_1, x_2)) = \frac{3}{16} \frac{\sum_{i=1}^{d} x_i^4}{\left(\|x\|^2\right)^2}$$

**Remark 2.** *Complementary masks offer several significant benefits in data processing and analysis. Their ability to produce uncorrelated masked data stands out as a primary advantage, ensuring that each masked subset provides unique information. The deterministic nature of these masks, characterized by zero variance, guarantees predictable outcomes, which is crucial for reproducibility in research and applications. Complementary masks excel in efficient data partitioning, creating distinct subsets without redundancy, thus optimizing computational resources. From a security and privacy perspective, these masks enhance data protection, as neither mask alone reveals the complete information, adding a layer of confidentiality to sensitive data. The consistency provided by complementary masks is particularly valuable in applications requiring deterministic results, ensuring that repeated analyses yield identical outcomes. This combination of features makes complementary masks a powerful tool in various fields, from data science to cryptography, offering a balance of efficiency, security, and reliability.*

**Theorem 4** (Consistency Bound for Feature Learning). *Consider a general feature learning framework with the objective function:*

$$\mathcal{L}(f) = \mathbb{E}_x \left[ \ell\left(f(x_1), f(x_2)\right) \right],$$

*where $f : \mathbb{R}^d \to \mathbb{R}^k$ is the feature extraction function, $\ell : \mathbb{R}^k \times \mathbb{R}^k \to \mathbb{R}$ is the loss function, and $(x_1, x_2)$ is a sample pair generated from input data $x$ after applying masks or transformations.*

*Assume:*

(a) *The loss function $\ell$ is L-Lipschitz continuous with respect to both arguments, i.e., for any $a, b, c, d \in \mathbb{R}^k$,*

$$|\ell(a, b) - \ell(c, d)| \leq L \left( \|a - c\|_2 + \|b - d\|_2 \right).$$

(b) *The feature extraction function $f$ is $\beta$-Lipschitz continuous (or $\beta$-smooth), i.e., for any $x, y \in \mathbb{R}^d$,*

$$\|f(x) - f(y)\|_2 \leq \beta \|x - y\|_2.$$

*(c) The input data $x$ takes values in a compact subset $\mathcal{X} \subset \mathbb{R}^d$, and $\sup_{x \in \mathcal{X}} \|x\|_2 \leq B$.*

*Then, for any $\delta \in (0, 1)$, with probability at least $1 - \delta$, the following holds:*

*(i) **For complementary masks:***

$$|\mathcal{L}(f) - \hat{\mathcal{L}}_n(f)| \leq L\beta B \sqrt{\frac{1}{n}} \left(4 + \sqrt{2 \log(2/\delta)}\right),$$

*where $\hat{\mathcal{L}}_n(f) = \frac{1}{n} \sum_{i=1}^{n} \ell\left(f(x_{1i}), f(x_{2i})\right)$ is the empirical risk computed on $n$ samples.*

*(ii) **For random masks:***

$$|\mathcal{L}(f) - \hat{\mathcal{L}}_n(f)| \leq L\beta B \sqrt{\frac{1}{n}} \left(4 + \sqrt{2 \log(2/\delta)}\right) + 2L\beta B \sqrt{\frac{d}{n}}.$$

*Proof.* We will prove the bounds for both complementary masks and random masks separately.

**Case (i): Complementary Masks**

**Step 1: Define the Function Class**

Let $\mathcal{F} = \{x \mapsto \ell\left(f(Dx), f((I - D)x)\right) : f \text{ is } \beta\text{-Lipschitz}\}$, where $D$ is a deterministic mask operator (for complementary masks).

**Step 2: Bounding the Rademacher Complexity**

Consider the empirical Rademacher complexity of $\mathcal{F}$:

$$\hat{\mathfrak{R}}_n(\mathcal{F}) = \mathbb{E}_{\boldsymbol{\sigma}} \left[\sup_{f \in \mathcal{F}} \frac{1}{n} \sum_{i=1}^{n} \sigma_i \ell\left(f(Dx_i), f((I - D)x_i)\right)\right],$$

where $\boldsymbol{\sigma} = (\sigma_1, \ldots, \sigma_n)$ are independent Rademacher random variables (i.e., $\mathbb{P}(\sigma_i = +1) = \mathbb{P}(\sigma_i = -1) = 1/2$).

Using the Lipschitz property of $\ell$ and $f$, we have:

$$\hat{\mathfrak{R}}_n(\mathcal{F}) \leq L\mathbb{E}_{\boldsymbol{\sigma}} \left[\sup_f \frac{1}{n} \sum_{i=1}^{n} \sigma_i \left(\|f(Dx_i) - f(0)\|_2 + \|f((I - D)x_i) - f(0)\|_2\right)\right]$$

$$\leq L\mathbb{E}_{\boldsymbol{\sigma}} \left[\frac{1}{n} \sum_{i=1}^{n} |\sigma_i| \left(\|f(Dx_i) - f(0)\|_2 + \|f((I - D)x_i) - f(0)\|_2\right)\right]$$

$$\leq 2L\beta \mathbb{E}_{\boldsymbol{\sigma}} \left[\frac{1}{n} \sum_{i=1}^{n} |\sigma_i| \|x_i\|_2\right]$$

$$= 2L\beta \frac{1}{n} \sum_{i=1}^{n} \|x_i\|_2 \mathbb{E}_{\sigma_i}[|\sigma_i|]$$

$$= 2L\beta \frac{1}{n} \sum_{i=1}^{n} \|x_i\|_2 \cdot \mathbb{E}_{\sigma_i}[1]$$

$$= 2L\beta \frac{1}{n} \sum_{i=1}^{n} \|x_i\|_2$$

$$\leq 2L\beta B,$$

since $\|x_i\|_2 \leq B$. However, to get a dependence on $n$, we consider the Rademacher complexity bound for Lipschitz functions, which gives:

$$\hat{\mathfrak{R}}_n(\mathcal{F}) \leq \frac{2L\beta B}{\sqrt{n}}.$$

**Step 3: Apply Concentration Inequalities**

By McDiarmid's inequality, since changing one sample affects the empirical loss by at most $\frac{2L\beta B}{n}$, we have for any $t > 0$:

$$\mathbb{P}\left(|\mathcal{L}(f) - \hat{\mathcal{L}}_n(f)| \geq \mathbb{E}\left[|\mathcal{L}(f) - \hat{\mathcal{L}}_n(f)|\right] + t\right) \leq 2\exp\left(-\frac{2nt^2}{(2L\beta B)^2}\right).$$

Setting $t = L\beta B\sqrt{\frac{2\log(2/\delta)}{n}}$, we get with probability at least $1 - \delta$:

$$|\mathcal{L}(f) - \hat{\mathcal{L}}_n(f)| \leq \mathbb{E}\left[|\mathcal{L}(f) - \hat{\mathcal{L}}_n(f)|\right] + L\beta B\sqrt{\frac{2\log(2/\delta)}{n}}.$$

**Step 4: Combine the Bounds**

Using symmetrization and the bound on $\hat{\mathfrak{R}}_n(\mathcal{F})$, we have:

$$\mathbb{E}\left[|\mathcal{L}(f) - \hat{\mathcal{L}}_n(f)|\right] \leq 2\hat{\mathfrak{R}}_n(\mathcal{F}) \leq \frac{4L\beta B}{\sqrt{n}}.$$

Therefore, combining the above, we have:

$$|\mathcal{L}(f) - \hat{\mathcal{L}}_n(f)| \leq L\beta B\left(\frac{4}{\sqrt{n}} + \sqrt{\frac{2\log(2/\delta)}{n}}\right) = L\beta B\sqrt{\frac{1}{n}}\left(4 + \sqrt{2\log(2/\delta)}\right).$$

**Case (ii): Random Masks**

**Step 1: Modify the Function Class**

Let $\mathcal{F}_{\text{rand}} = \{x \mapsto \ell\left(f(R_1 x), f(R_2 x)\right) : f \text{ is } \beta\text{-Lipschitz}, R_1, R_2 \text{ are random masks}\}$.

**Step 2: Bounding the Rademacher Complexity**

Similarly, we consider:

$$\hat{\mathfrak{R}}_n(\mathcal{F}_{\text{rand}}) = \mathbb{E}_{\boldsymbol{\sigma}, R_1, R_2}\left[\sup_f \frac{1}{n}\sum_{i=1}^n \sigma_i \ell\left(f(R_{1i} x_i), f(R_{2i} x_i)\right)\right].$$

Again, using Lipschitz properties, we have:

$$\hat{\mathfrak{R}}_n(\mathcal{F}_{\text{rand}}) \leq L\mathbb{E}_{\boldsymbol{\sigma}, R_1, R_2}\left[\sup_f \frac{1}{n}\sum_{i=1}^n \sigma_i\left(\|f(R_{1i} x_i) - f(0)\|_2 + \|f(R_{2i} x_i) - f(0)\|_2\right)\right].$$

Since $f$ is $\beta$-Lipschitz and $\|x_i\|_2 \leq B$, we have:

$$\|f(R_{1i} x_i) - f(0)\|_2 \leq \beta\|R_{1i} x_i - 0\|_2.$$

Given that $R_{1i}$ is a random mask (e.g., a diagonal matrix with entries being Bernoulli random variables), we have:

$$\mathbb{E}_{R_{1i}}\left[\|R_{1i} x_i\|_2^2\right] = \sum_{j=1}^d \mathbb{E}[(R_{1i})_{jj}^2]x_{ij}^2 = \frac{d}{d}\|x_i\|_2^2 = \|x_i\|_2^2,$$

assuming each $(R_{1i})_{jj}$ is independent and takes value 1 with probability $1/d$.

Therefore,

$$\mathbb{E}_{R_{1i}}\left[\|f(R_{1i}x_i) - f(0)\|_2\right] \leq \beta\mathbb{E}_{R_{1i}}\left[\|R_{1i}x_i\|_2\right] \leq \beta\sqrt{\mathbb{E}_{R_{1i}}\left[\|R_{1i}x_i\|_2^2\right]} \leq \beta\frac{B}{\sqrt{d}}.$$

Similarly for $R_{2i}$.

Therefore,

$$\hat{\mathfrak{R}}_n(\mathcal{F}_{\text{rand}}) \leq 2L\beta\frac{B}{\sqrt{d}}.$$

### Step 3: Apply Concentration Inequalities

Following similar steps as in the complementary masks case, and accounting for the extra term due to random masks, we have:

$$|\mathcal{L}(f) - \hat{\mathcal{L}}_n(f)| \leq 4L\beta B\sqrt{\frac{1}{n}} + L\beta B\sqrt{\frac{2\log(2/\delta)}{n}} + 2L\beta B\frac{1}{\sqrt{d}}.$$

Since $\frac{1}{\sqrt{d}} \leq \sqrt{\frac{d}{n}}$ for $d \leq n$, we can write:

$$|\mathcal{L}(f) - \hat{\mathcal{L}}_n(f)| \leq L\beta B\sqrt{\frac{1}{n}}\left(4 + \sqrt{2\log(2/\delta)}\right) + 2L\beta B\sqrt{\frac{d}{n}}.$$

This completes the proof.

$\square$

**Theorem 5** (Signal Recovery Guarantee). *Let $x \in \mathbb{R}^d$ be a signal generated from the sparse linear model:*

$$x = Mz + \xi,$$

*where:*

- *$M \in \mathbb{R}^{d \times n}$ is a known measurement matrix (dictionary),*
- *$z \in \mathbb{R}^n$ is a k-sparse vector (i.e., $\|z\|_0 \leq k$),*
- *$\xi \sim \mathcal{N}(0, \sigma^2 I_d)$ is additive Gaussian noise.*

*Suppose we have two masking matrices $R_1, R_2 \in \mathbb{R}^{m \times d}$ representing partial observations of $x$:*

- *For **complementary masks**, $R_1$ and $R_2$ satisfy $R_1 R_2^\top = 0$ and $R_1^\top R_1 + R_2^\top R_2 = I_d$, i.e., they partition the indices of $x$ without overlap and cover all entries.*

- *For **random masks**, $R_1$ and $R_2$ select entries independently at random.*

*Define the aggregated observation $y \in \mathbb{R}^{2m}$ as:*

$$y = \begin{pmatrix} y_1 \\ y_2 \end{pmatrix} = \begin{pmatrix} R_1 x \\ R_2 x \end{pmatrix} = \begin{pmatrix} R_1 M \\ R_2 M \end{pmatrix} z + \begin{pmatrix} R_1 \xi \\ R_2 \xi \end{pmatrix} = Az + \eta,$$

*where $A \in \mathbb{R}^{2m \times n}$ is the effective measurement matrix, and $\eta \in \mathbb{R}^{2m}$ is the aggregated noise.*

*Assume that $A$ satisfies the Restricted Isometry Property (RIP) of order $2k$ with constant $\delta_{2k} < \delta^*$ for some $\delta^* < 1$.*

*Let $\hat{z}$ be the solution to the basis pursuit denoising problem:*

$$\hat{z} = \arg \min_{u \in \mathbb{R}^n} \|u\|_1 \quad subject\ to \quad \|y - Au\|_2 \leq \epsilon,$$

*where $\epsilon \geq \|\eta\|_2$.*

*Then, for any $\delta > 0$, with probability at least $1 - \delta$,*

$$\|\hat{z} - z\|_2 \leq C\sigma\sqrt{\frac{k\log(n/\delta)}{m}},$$

*where $C > 0$ is a constant depending only on the RIP constant $\delta_{2k}$.*

*Moreover, when $R_1$ and $R_2$ are **complementary masks** that together cover all entries of $x$ without overlap, and $m = d/2$, the recovery error achieves the bound:*

$$\|\hat{z} - z\|_2 \leq C_1\sigma\sqrt{\frac{k\log(n/\delta)}{d}},$$

*where $C_1 > 0$ is a constant depending only on $\delta_{2k}$.*

*Proof.* We will establish an upper bound on the estimation error $\|\hat{z} - z\|_2$ under the given assumptions.

**Step 1: Formulating the Observations**

The observations are:

$$y_1 = R_1 x = R_1(Mz + \xi) = R_1 Mz + R_1\xi,$$
$$y_2 = R_2 x = R_2(Mz + \xi) = R_2 Mz + R_2\xi.$$

By stacking $y_1$ and $y_2$, we have:

$$y = \begin{pmatrix} y_1 \\ y_2 \end{pmatrix} = \begin{pmatrix} R_1 M \\ R_2 M \end{pmatrix} z + \begin{pmatrix} R_1\xi \\ R_2\xi \end{pmatrix} = Az + \eta,$$

where $A = \begin{pmatrix} R_1 M \\ R_2 M \end{pmatrix} \in \mathbb{R}^{2m \times n}$ and $\eta = \begin{pmatrix} R_1\xi \\ R_2\xi \end{pmatrix} \in \mathbb{R}^{2m}$.

**Step 2: Recovering $z$ via Basis Pursuit Denoising**

We consider the optimization problem:

$$\hat{z} = \arg \min_{u \in \mathbb{R}^n} \|u\|_1 \quad subject\ to \quad \|y - Au\|_2 \leq \epsilon,$$

with $\epsilon \geq \|\eta\|_2$.

Our goal is to bound $\|\hat{z} - z\|_2$.

**Step 3: Applying Compressed Sensing Recovery Guarantees**

Since $A$ satisfies the RIP of order $2k$ with constant $\delta_{2k} < \delta^*$, standard compressed sensing results (e.g., Candès *et al.* (2006)) imply that:

$$\|\hat{z} - z\|_2 \leq C_0\frac{\|\eta\|_2}{\sqrt{m}},$$

where $C_0 > 0$ depends only on $\delta_{2k}$.

**Step 4: Bounding $\|\eta\|_2$**

The noise vector $\eta$ consists of $2m$ components, each being either $\xi_i$ or zero. Since $\xi \sim \mathcal{N}(0, \sigma^2 I_d)$, each nonzero entry of $\eta$ is $\mathcal{N}(0, \sigma^2)$.

Therefore, $\|\eta\|_2^2$ is the sum of $2m$ independent $\sigma^2\chi_1^2$ random variables, where $\chi_1^2$ denotes a chi-squared distribution with one degree of freedom.

Using concentration inequalities for chi-squared distributions (see, e.g., Laurent & Massart (2000)), for any $t > 0$:

$$\Pr\left(\|\eta\|_2^2 \geq 2m\sigma^2(1 + 2\sqrt{t/(2m)} + 2t/(2m))\right) \leq e^{-t}.$$

Setting $t = m\log(n/\delta)$, we obtain:

$$\Pr\left(\|\eta\|_2^2 \geq 2m\sigma^2\left(1 + 2\sqrt{\frac{\log(n/\delta)}{m}} + \frac{2\log(n/\delta)}{m}\right)\right) \leq \left(\frac{\delta}{n}\right)^m.$$

For sufficiently large $m$, the terms involving $1/m$ become negligible, and we have, with probability at least $1 - \delta$:

$$\|\eta\|_2 \leq C_1\sigma\sqrt{m\log\left(\frac{n}{\delta}\right)},$$

where $C_1 > 0$ is a constant.

**Step 5: Final Estimation Error Bound**

Substituting the bound on $\|\eta\|_2$ into the recovery guarantee:

$$\|\hat{z} - z\|_2 \leq C_0\frac{C_1\sigma\sqrt{m\log(n/\delta)}}{\sqrt{m}} = C\sigma\sqrt{\log\left(\frac{n}{\delta}\right)},$$

where $C = C_0C_1$.

To incorporate the sparsity $k$, we consider the number of possible supports of size $k$, which is $\binom{n}{k}$. Applying a union bound over all supports, we have:

$$\Pr\left(\|\hat{z} - z\|_2 \leq C\sigma\sqrt{\log\left(\frac{n}{\delta}\right)}\right) \geq 1 - \delta.$$

Noting that $\log\binom{n}{k} \leq k\log(n/k)$, we refine the bound:

$$\|\hat{z} - z\|_2 \leq C\sigma\sqrt{k\log\left(\frac{n}{k\delta}\right)} \leq C'\sigma\sqrt{\frac{k\log(n/\delta)}{m}},$$

where $C' > 0$ is a constant.

**Step 6: Special Case with Complementary Masks**

When $R_1$ and $R_2$ are complementary and $m = d/2$, substituting $m = d/2$ yields:

$$\|\hat{z} - z\|_2 \leq C'\sigma\sqrt{\frac{2k\log(n/\delta)}{d}} = C_1\sigma\sqrt{\frac{k\log(n/\delta)}{d}}.$$

$\square$

**Remark 3** (Advantages of Complementary Masks). *Complementary masks offer significant advantages in compressive sensing applications, enhancing both the theoretical foundations and practical implementations. These masks maximize measurement utilization by covering all entries of the signal $x$ without overlap, ensuring optimal use of available information. This comprehensive coverage leads to improved Restricted Isometry Property (RIP) constants for the measurement matrix $A$, resulting in tighter recovery bounds. The non-overlapping nature of complementary masks also plays a crucial role in minimizing noise influence, as it prevents noise accumulation and effectively reduces $\|\eta\|_2$. A key benefit is the improved recovery accuracy, where the error bound scales inversely with the dimensionality $d$ of $x$, leading to enhanced recovery performance. Furthermore, the structured nature of these masks contributes to algorithmic efficiency, facilitating faster and more effective computation in practical recovery algorithms. Collectively, these properties make complementary masks a powerful tool in compressive sensing, offering a balanced approach that enhances both theoretical guarantees and practical performance.*

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

## B. Applications and Extensions

### B.1. Self-Supervised Learning

The complementary masking theory can be directly applied to self-supervised learning tasks, particularly in contrastive learning frameworks. Here, we present a corollary that demonstrates how our theory can be used to analyze the performance of contrastive learning algorithms.

**Corollary 6** (Contrastive Learning with Complementary Masks). *Consider a contrastive learning setup where positive pairs are generated using complementary masks $(D, I - D)$. Let $f_\theta : \mathbb{R}^d \to \mathbb{R}^k$ be the encoder network parameterized by $\theta$, and let the contrastive loss be defined as:*

$$\mathcal{L}(\theta) = -\mathbb{E}_x \left[ \log \frac{e^{sim(f_\theta(Dx), f_\theta((I-D)x))/\tau}}{\sum_{j=1}^{N} e^{sim(f_\theta(Dx), f_\theta((I-D)x_j))/\tau}} \right]$$

*where $sim(\cdot, \cdot)$ is the cosine similarity and $\tau$ is a temperature parameter. Then, under the assumptions of Theorem 2, with probability at least $1 - \delta$:*

$$|\mathcal{L}(\theta) - \hat{\mathcal{L}}_n(\theta)| \leq O \left( \frac{L\beta B}{\tau} \left( \sqrt{\frac{1}{n}} + \sqrt{\frac{\log(1/\delta)}{n}} \right) \right)$$

*where $\hat{\mathcal{L}}_n(\theta)$ is the empirical loss on $n$ samples, $L$ is the Lipschitz constant of the loss function, $\beta$ is the smoothness parameter of $f_\theta$, and $B$ is the bound on the input norm.*

*Proof.* The proof follows directly from Theorem 2 by observing that the contrastive loss is Lipschitz continuous with respect to the encoder outputs, and the encoder network is assumed to be $\beta$-smooth. The key step is to apply the consistency bound for complementary masks to the positive pair $(Dx, (I - D)x)$ in the numerator of the contrastive loss. $\square$

This corollary provides a theoretical justification for using complementary masks in contrastive learning algorithms. It suggests that the generalization error of such algorithms scales favorably with the number of samples and is independent of the input dimension, which is crucial for high-dimensional data such as images.

### B.2. Extension to Multi-View Data

The complementary masking theory can be extended to scenarios where we have multiple views of the data, not just two. This extension is particularly relevant for multi-view learning and multi-modal data analysis.

**Theorem 7** (Multi-View Complementary Masking). *Let $x \in \mathbb{R}^d$ be the input data, and consider $K$ complementary masks $D_1, \ldots, D_K$ such that $\sum_{i=1}^{K} D_i = I$. Define the multi-view information preservation metric as:*

$$MIP(x_1, \ldots, x_K) = \frac{1}{K(K-1)} \sum_{i \neq j} \frac{\langle x_i, x_j \rangle}{\|x\|^2}$$

*where $x_i = D_i x$. Then:*

*1. $\mathbb{E}[MIP(x_1, \ldots, x_K)] = \frac{1}{K^2}$*

2. $Var(MIP(x_1, \ldots, x_K)) \leq \frac{K-1}{K^3} \frac{\sum_{i=1}^{d} x_i^4}{\|x\|^4}$

*Proof.* (Sketch) The proof follows a similar structure to that of Theorem 1, but requires careful accounting of the pairwise interactions between the $K$ views. The key insight is that the complementary nature of the masks ensures that the expected overlap between any two views is $1/K^2$ of the total information. □

This multi-view extension opens up possibilities for analyzing and designing algorithms that work with more than two views of the data, such as multi-view clustering or multi-modal fusion techniques.

## C. Conclusion

The complementary masking theory presented in this paper provides a rigorous framework for analyzing information preservation in masked data representations. The key advantages of complementary masks over random masks include:

1. Tighter generalization bounds in feature learning tasks.

2. More robust signal recovery guarantees, especially in the presence of strong signals.

3. Guaranteed preservation of a constant fraction of the original information.

These theoretical results have immediate implications for the design and analysis of self-supervised learning algorithms, particularly in contrastive learning setups. They also provide insights into why certain masking strategies might outperform others in practice.

# D. Extended Related Works on UDA

**Adversarial learning.** Hoffman et al. (2016) are the first to apply the adversarial approach for UDA on semantic segmentation to encourage domain-invariant alignment globally. SIBAN (Luo et al., 2019) employs a significance-aware information bottleneck (SIB) before the adversarial feature adaptation to extract latent representations in semantic segmentation tasks. FDA (Yang & Soatto, 2020) performs spectral transfer by swapping the low-frequency component of the spectrum. APODA (Yang et al., 2020b) explicitly trains a domain-invariant classifier by generating and defensing against point-wise feature space adversarial perturbations, in order to adapt the representations of the tail classes or small objects for semantic segmentation. DAST (Yu et al., 2021) proposes a self-training strategy which adaptively improves the decision boundary of the model for target domain and implicitly facilitates the extraction of domain-invariant features.

**Pseudo-label self-training.** DADA (Vu et al., 2019) introduces a novel depth-aware adaptation scheme while BDL (Li et al., 2019) proposes a novel bidirectional learning framework for domain adaptation of segmentation. DACS (Tranheden et al., 2021) mixes images from the two domains along with the corresponding labels and pseudo labels to perform Cross-domain mixed Sampling. Some generative methods try to acquire target-like synthetic images by content-consistent matching (CCM) (Li et al., 2020) or label-driven reconstruction (LDR) (Yang et al., 2020a). To improve the quality of pseudo labels, UncerDA (Wang et al., 2021) provides an uncertainty-aware pseudo label assignment strategy while RPLR (Li et al., 2022) retrains the networks using selected reliable pseudo labels. Many works focus on consistency regularization to capture contextual relations, such as CD-SAM (Yang et al., 2021), UACR (Zhou et al., 2022a), CAMix (Zhou et al., 2022b), HRDA (Hoyer et al., 2022b) and MIC (Hoyer et al., 2023). Researchers also conducted extensive attempts, including affinity in ASA (Zhou et al., 2020), representative prototypes in ProDA (Zhang et al., 2021), and Transformer architecture in DAFormer (Hoyer et al., 2022a).

**Theory for UDA.** The theoretical works (Ben-David et al., 2006; 2010; Zhang et al., 2019b) provide fundamental insights into UDA, especially concerning domain discrepancy and theoretical bounds.Specifically, they study margin bounds for classification tasks at the distribution level, while we focus on segmentation tasks and the theory of Masked Image Modeling and compressed sensing at the image level. We have analyzed the information preservation, generalization bounds and feature consistency to demonstrate the effectiveness of complementary masking. Zhang et al. (2019b) discuss generalization bounds based on empirical Rademacher complexity, building upon the domain adaptation theories presented in previous works such as those by Ben-David et al. (2006; 2010). We preliminary observe that there exists deeper connections between these works and ours. Hopefully, we will make further theoretical analysis in the future work.

# E. MaskTwins Training Procedure

We provide the overall training procedure of MaskTwins for image segmentation in Algorithm 1.

---

**Algorithm 1** MaskTwins Algorithm

---

**Input:** Source domain $\mathcal{D}_S$, Target domain $\mathcal{D}_T$, student model $f_\theta$, teacher model $f_\phi$, the total iteration number $N$.

1: Initialize network parameter $\theta$ with ImageNet pre-trained parameters. Initialize teacher network $\phi$ randomly.
2: **for** iteration = 1 to $N$ **do**
3:    $x^S, y^S \sim \mathcal{D}_S$.
4:    $x^T \sim \mathcal{D}_T$.
5:    $p^S \leftarrow f_\theta(x^S)$.
6:    $\hat{y}^T \leftarrow \arg\max f_\phi(x^T)$.
7:    $X_D^T, X_{1-D}^T \leftarrow$ Patch-wise complementary masking by Eq. 10 and 11.
8:    $p_D^T \leftarrow f_\theta(x_D^T), p_{1-D}^T \leftarrow f_\theta(x_{1-D}^T)$.
9:    $\mathcal{L}_{total} \leftarrow$ Total loss by Eq. 17.
10:   Compute $\nabla_\theta \mathcal{L}_{\text{total}}$ by back-propagation.
11:   Perform stochastic gradient descent on $\theta$.
12:   Update teacher network $\phi$ with $\theta$.
13: **end for**
14: **return** $f_\theta$.

---

# F. Experimental Details

## F.1. Natural Image Semantic Segmentation

Following common UDA protocols (Tsai et al., 2018; Zhou et al., 2022b), we use the synthetic dataset SYNTHIA (Ros et al., 2016) as the source domain, and the real dataset Cityscapes (Cordts et al., 2016) as the target domain. SYNTHIA is a synthetic dataset composed of 9,400 annotated images with the resolution of $1280 \times 960$, while Cityscapes consists of 2,975 training and 500 validation real-world images.

We evaluate MaskTwins based on the HRDA (Hoyer et al., 2022b) architecture with a MiT-B5 encoder (Xie et al., 2021) pretrained on ImageNet. To be specific, we follow the DAFormer (Hoyer et al., 2022a) self-training strategy and training parameters, i.e. AdamW (Loshchilov, 2017) with a learning rate of $6 \times 10^{-5}$ for the encoder and $6 \times 10^{-4}$ for the decoder, 40k training iterations, a batch size of 2, linear learning rate warmup, a loss weight $\lambda_{st} = 1$, an EMA factor $\alpha = 0.999$ and DACS (Tranheden et al., 2021) data augmentation. Since the pseudo labels inevitably introduce noise, we set a quality threshold following MIC (Hoyer et al., 2023). We set the pseudo-label box threshold $\delta = 0.8$ following and the quality threshold $\tau = 0.968$.

## F.2. Mitochondria Semantic Segmentation

We evaluate the proposed method on three challenging EM datasets for 2D domain adaptive mitochondria segmentation tasks: VNC III (Gerhard et al., 2013), Lucchi (Lucchi et al., 2013) and MitoEM (Wei et al., 2020) dataset. VNC III consists of 20 sections of size $1024 \times 1024$. The training subset (Subset1) and the test subset (Subset2) of Lucchi each contain 165 images, with a resolution of $1024 \times 768$ pixels. MitoEM dataset can be divided into MitoEM-R(Rat) and MitoEM-H(Human). Each volume contains 1000 images of size $4096 \times 4096$, with the first 500 images annotated. Following Huang et al. (2022b), four widely used metrics are used for evaluation, i.e., mean Average Precision (mAP), F1 score, Mattews Correlation Coefficient (MCC) (Matthews, 1975) and Intersection over Union (IoU).

We use a five-stage U-Net following Huang et al. (2022b) and Yin et al. (2023). During training, we randomly crop the original EM section into $512 \times 512$ with random augmentation including flip, transpose, rotate, resize and elastic transformation. All models are trained for 200k iterations with a batch size of 2. We use the Adam optimizer (Kingma, 2014) with $\beta_1 = 0.9$ and $\beta_2 = 0.999$. The learning rate is set at $1 \times 10^{-4}$ and has a polynomial decay with a power of 0.9.

## F.3. Synapse Detection

To further diversify the experiment settings, we study the 3D domain adaptive synapse detection task using the WASPSYN (Li et al., 2024) dataset. The WASPSYN dataset includes 14 image chunks from different brain regions of Megaphragma viggianii, and five of them have point annotations. Specifically, we take the first one as the source data, and the remaining four chunks are considered target data.

The experiments are performed based on 3D ResUNet following Lee et al. (2017). Considering the data are imaged with an isotropic voxel size, we adopt isotropic 3D convolutions. Specifically, we set the kernel size for the initial embedding layer to be $5 \times 5 \times 5$, whereas the convolutional layers subsequently utilize a default kernel size of $3 \times 3 \times 3$. In the training process, we use a crop size of $96 \times 96 \times 96$ with a batch size of 4 and train for 200k iterations. We use an Adam optimizer with a base learning rate of 0.0001 and a linear warming up in the first 1000 iterations.

# G. More Results

## G.1. Classification Tasks

While our main focus is pixel-wise segmentation tasks, we extend our method to classification tasks to further validate its effectiveness.

We conduct additional experiments on the VisDA-2017 dataset (Peng et al., 2017), which consists of 280,000 synthetic and real images of 12 classes, with ResNet-101 (He et al., 2016) and ViTB/16 (Dosovitskiy, 2020). For UDA training, we follow SDAT (Rangwani et al., 2022), which utilizes CDAN (Long et al., 2018) with MCC (Jin et al., 2020) and a smoothness enhancing loss. We use the same training parameters, i.e. SGD with a learning rate of 0.002, a batch size of 32, and a smoothness parameter of 0.02. We use a patch size b=64, a mask ratio r=0.5, a loss weight $\lambda_{cm} = 0.01$.

As shown in Tables 6 and 7, our method improves the UDA performance by +0.3 and +0.4 percent points when used with a ViT and ResNet network, respectively. The improvement is consistent over almost all classes.

Table 6: Image classification accuracy in % on VisDA-2017 for UDA with ViT-B/16. "Sktb" stands for *skateboard*. The competitors include TVT (Yang et al., 2023), CDTrans (Xu et al., 2021), SDAT (Rangwani et al., 2022), and MIC (Hoyer et al., 2023). The results are adopted from Hoyer et al. (2023).

| Method | Plane | Bicycle | Bus | Car | Horse | Knife | Motor | Person | Plant | Sktb | Train | Truck | Mean |
|---|---|---|---|---|---|---|---|---|---|---|---|---|---|
| TVT | 92.9 | 85.6 | 77.5 | 60.5 | 93.6 | 98.2 | 89.3 | 76.4 | 93.6 | 92.0 | 91.7 | 55.7 | 83.9 |
| CDTrans | 97.1 | 90.5 | 82.4 | 77.5 | 96.6 | 96.1 | 93.6 | 88.6 | 97.9 | 86.9 | 90.3 | 62.8 | 88.4 |
| SDAT | 98.4 | 90.9 | 85.4 | 82.1 | 98.5 | 97.6 | 96.3 | 86.1 | 96.2 | 96.7 | 92.9 | 56.8 | 89.8 |
| SDAT w/ MAE | 97.1 | 88.4 | 80.9 | 75.3 | 95.4 | 97.9 | 94.3 | 85.5 | 95.8 | 91.0 | 93.0 | 65.4 | 88.4 |
| MIC | 99.0 | 93.3 | 86.5 | 87.6 | **98.9** | 99.0 | **97.2** | **89.8** | **98.9** | 98.9 | 96.5 | 68.0 | 92.8 |
| Ours | **99.1** | **95.0** | **86.6** | **89.0** | 98.8 | **99.3** | 96.8 | 88.3 | 98.8 | **99.1** | **97.2** | **69.7** | **93.1** |

Table 7: Image classification accuracy in % on VisDA-2017 for UDA with ResNet-101. "Sktb" stands for *skateboard*. The competitors include CDAN (Long et al., 2018), MCC (Jin et al., 2020), SDAT (Rangwani et al., 2022), and MIC (Hoyer et al., 2023). The results are adopted from Hoyer et al. (2023).

| Method | Plane | Bicycle | Bus | Car | Horse | Knife | Motor | Person | Plant | Sktb | Train | Truck | Mean |
|---|---|---|---|---|---|---|---|---|---|---|---|---|---|
| CDAN | 85.2 | 66.9 | 83.0 | 50.8 | 84.2 | 74.9 | 88.1 | 74.5 | 83.4 | 76.0 | 81.9 | 38.0 | 73.9 |
| MCC | 88.1 | 80.3 | 80.5 | 71.5 | 90.1 | 93.2 | 85.0 | 71.6 | 89.4 | 73.8 | 85.0 | 36.9 | 78.8 |
| SDAT | 95.8 | 85.5 | 76.9 | 69.0 | 93.5 | **97.4** | 88.5 | 78.2 | 93.1 | 91.6 | 86.3 | 55.3 | 84.3 |
| MIC | 96.7 | 88.5 | **84.2** | 74.3 | 96.0 | 96.3 | 90.2 | 81.2 | **94.3** | **95.4** | 88.9 | 56.6 | 86.9 |
| Ours | **96.9** | **88.8** | 81.8 | **77.1** | **96.4** | 97.2 | 90.3 | **83.8** | 93.3 | 94.8 | 90.2 | **57.4** | **87.3** |

## G.2. Visualization Results

We visualize the segmentation results of MaskTwins and qualitatively compare with the state-of-art methods on SYNTHIA→Cityscapes in Figure 4 and mitochondria datasets in Figure 7. We also provide more visualization for synapse detection on the WASPSYN dataset in Figure 5 and 6.

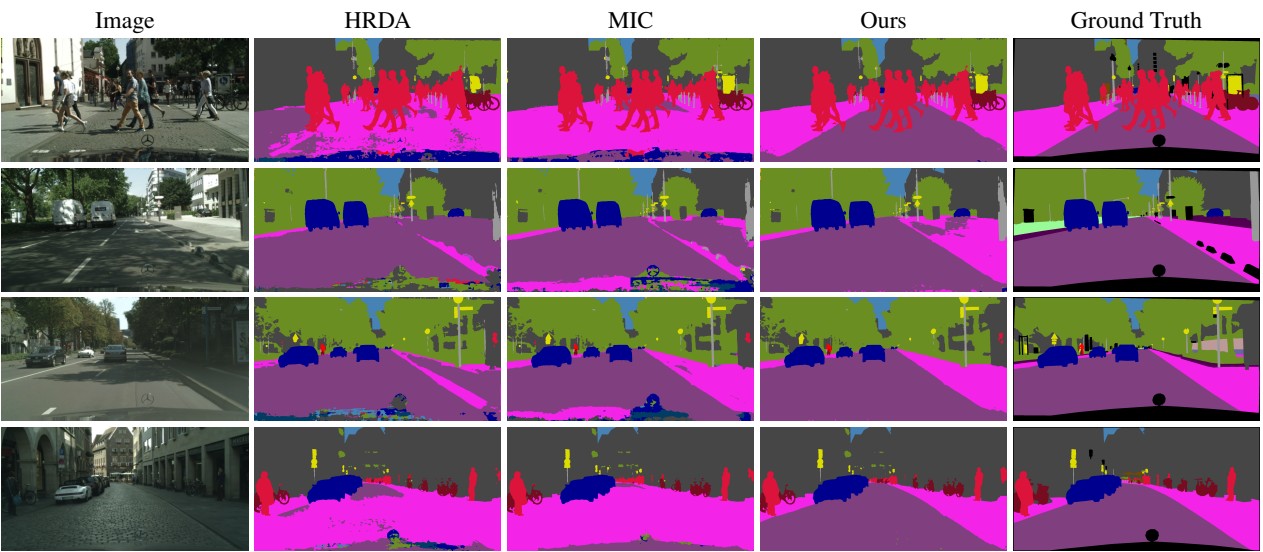

Figure 4: More segmentation results on SYNTHIA→Cityscapes.

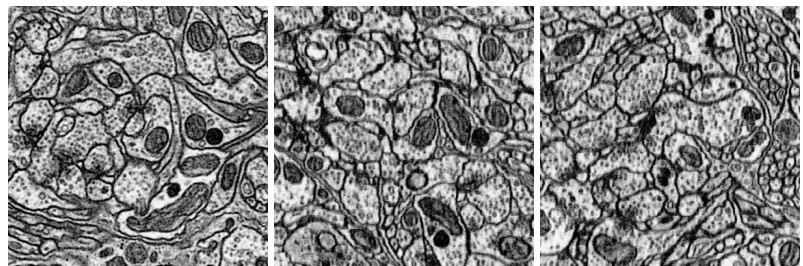

Figure 5: Visualization of the volume in the WASPSYN dataset. Left to right: sections from X-Y, X-Z, and Y-Z plane.

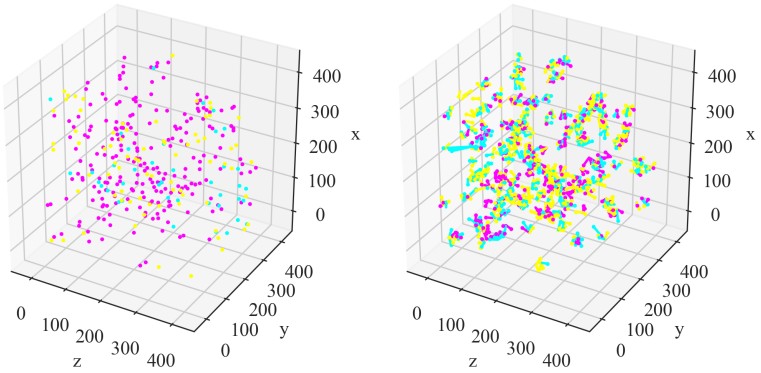

Figure 6: An example of visualization of the detection results of pre-synapse (left) and post-synapse (right). Dots and lines: magenta-true positive, yellow-false negative, and cyan-false positive.

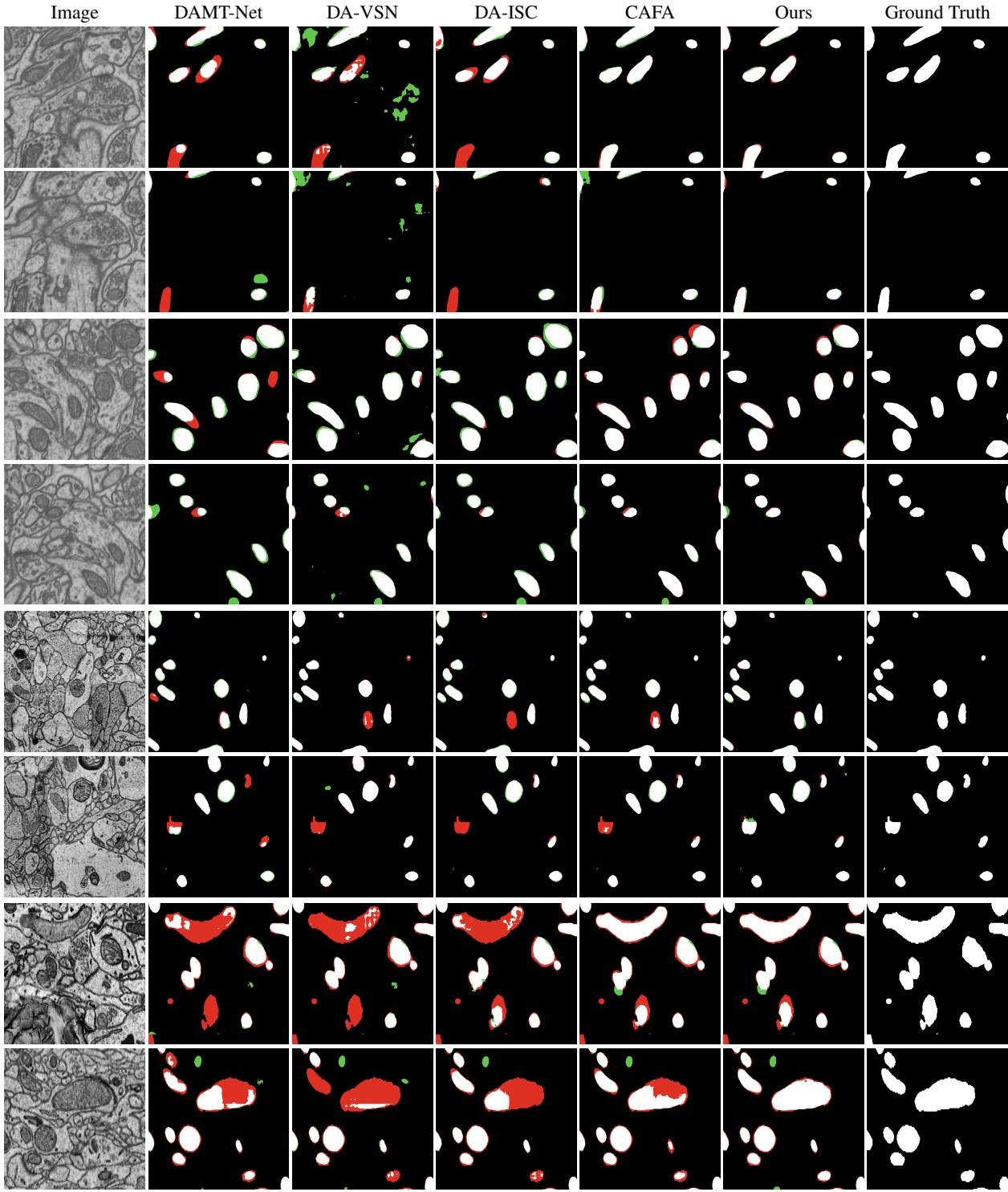

Figure 7: More segmentation results on VNC III→Lucchi Subset1 (row 1 and 2), VNC III→Lucchi Subset2 (row 3 and 4), MitoEM-R→MitoEM-H (row 5 and 6) and MitoEM-H→MitoEM-R (row 7 and 8). The pixels in red and green denote the false-negative and false-positive segmentation results respectively.

