# OpenReview forum: "MaskTwins: Dual-form Complementary Masking for Domain-Adaptive Image Segmentation"
_ICML.cc/2025/Conference — ICML 2025 poster_

### Official Review · Reviewer_7ChN · 2025-03-10

**Overall Recommendation:** 3

**Summary:**

This paper tackles image segmentation within the unsupervised domain adaptation (UDA) framework. Rather than employing random masks, the authors develop dual-form complementary masked images to strengthen the generalization capabilities of their approach. They demonstrate that robust, domain-invariant features can be extracted through consistency learning applied to these masked images. The method's effectiveness is validated through comprehensive experiments across six distinct test cases.

**Claims And Evidence:**

To comprehensively understand the effectiveness of the proposed Complementary Mask, I recommend the authors can conduct experiments on some more challenging segmentation datasets. For example, the medical image segmentation has strong domain shifts across domains. The fundus segmentation tasks such as OC and OD segmentation is a good choice.

**Essential References Not Discussed:**

N/A

**Experimental Designs Or Analyses:**

I appreciate nice numerical results, but it is also necessary for the reader to vividly understand why the proposed Complementary Mask works. I suggest some feature visualizations may be added to compare with random mask scheme.

**Methods And Evaluation Criteria:**

Since the overall framework of this paper relies on multiple component e.g., the AdaIN and the EMA-based pseudo labels, it seems to be over-complicated and is not elegant enough from a perspective of ML community. Meanwhile, according to the ablation study, it seems that the proposed Complementary Mask cannot work on Complementary Mask-only framework? This eliminates the effectiveness of the proposed Complementary Mask. I suggest the authors can report the result of a Complementary Mask-only framework to demonstrate the efficacy of the proposed Complementary Mask framework.

**Other Comments Or Suggestions:**

Please see weakness

**Other Strengths And Weaknesses:**

Strengths:

- This paper introduces the Complementary Mask strategy based on random mask, the empirical experiments look good.
- This paper is easy to follow and well-written.
- The authors provide theoretical analysis to justify the proposed complementary mask related to the sparse signal reconstruction.

Weakness:

- Since the overall framework of this paper relies on multiple component e.g., the AdaIN and the EMA-based pseudo labels, it seems to be over-complicated and is not elegant enough from a perspective of ML community. Meanwhile, according to the ablation study, it seems that the proposed Complementary Mask cannot work on Complementary Mask-only framework? This eliminates the effectiveness of the proposed Complementary Mask. I suggest the authors can report the result of a Complementary Mask-only framework to demonstrate the efficacy of the proposed Complementary Mask framework.

- To be honest, although I try to understand the envisioned Theorems, I find that there is not a directly related connection between the sparse signal reconstruction in Assumption 1 and corresponding generalization bound as well as feature consistency. Therefore, I do not know how the actually sparse signal reconstruction theory is utilized? It seems that envision Theorems are not directly link to the claimed contribution 1.

- To comprehensively understand the effectiveness of the proposed Complementary Mask, I recommend the authors can conduct experiments on some more challenging segmentation datasets. For example, the medical image segmentation has strong domain shifts across domains. The fundus segmentation tasks such as OC and OD segmentation is a good choice.

- I appreciate nice numerical results, but it is also necessary for the reader to vividly understand why the proposed Complementary Mask works. I suggest some feature visualizations may be added to compare with random mask scheme.

=====================================================================================
Thanks for authors' responses that have addressed my concerns. I have raised my score.

**Questions For Authors:**

Please see weaknesses

**Relation To Broader Scientific Literature:**

N/A

**Theoretical Claims:**

To be honest, although I try to understand the envisioned Theorems, I find that there is not a directly related connection between the sparse signal reconstruction in Assumption 1 and corresponding generalization bound as well as feature consistency. Therefore, I do not know how the actually sparse signal reconstruction theory is utilized? It seems that envision Theorems are not directly link to the claimed contribution 1.

---

> ### Author Rebuttal · Authors · 2025-04-01
>
> Thank you for your valuable time and comments. The main concerns are addressed below.
>
> > **W1:** Complementary Mask-only framework
>
> The proposed complementary masking strategy certainly works on the Complementary Mask-only framework and we highlight that it does not rely on the network structure or additional internal components.
>
> Specifically, AdaIN is a classic component used in domain adaptation and EMA commonly serves for teacher-student structure. We adopt the EMA teacher model for pseudo-label generation, following common practices in UDA methods [1]. These components are simple and require no special hyper-parameter tuning. In fact, the corresponding contributions are quite limited according to Table 4.
>
> Considering the misunderstanding caused by, we have supplemented Table 4 with the results of Complementary Mask-only framework as follows. The results show that remarkable gains can be achieved by using the complementary mask alone. We highly agree that this can better demonstrate the effectiveness of the proposed complementary mask framework, and we will include this in the revised paper.
>
> ---
>
> Table R1. Supplementary ablation study based on Table 4.
>
> |  CL  | CMask | RMask | EMA  | AdaIN | mIoU |
> | :--: | :---: | :---: | :--: | :---: | :--: |
> |  √   |   √   |   -   |  -   |   -   | 76.0 |
> ---
>
> > **W2:** Question on theorem
>
> The assumption of sparse signal in Assumption 1 is a general background.  This assumption serves as the foundation for our theoretical analysis. Although we use the image (X) in the formula without decomposing it into the sparse signal (S) and the noise (N), it holds true completely in both the analysis of the generalization bound and that of the feature consistency. This connection is implicit and more importantly, intrinsic. In our analysis, the generalization bound focuses on the theoretical explanation of mask reconstruction, while feature consistency is applied to account for the setting of domain adaptation.
>
> Meanwhile, we explicitly use Assumption 1 in the proofs of Theorem 5 (Signal Recovery Guarantee) in the supplementary materials. Specifically, we reframe X as a signal generated from the sparse linear model, and apply Compressed Sensing Recovery Guarantees during the proofs.
>
> > **W3:** Application on more challenging segmentation dataset
>
> While our main experiments have comprehensively covered both natural images and medical images, we agree that extending our method to fundus segmentation tasks can further validate its effectiveness. To address this, we conducted additional experiments on the RIM-ONE-r3 dataset.
>
> ---
>
> Table R2. Quantitative results of OD and OC segmentation on the RIM-ONE-r3 dataset with a CBMT backbone, using only the proposed complementary masking strategy.
>
> |            | $Dice_{OD}$↑  |  $ASSD_{OD}$↓   | $Dice_{OC}$↑  |  $ASSD_{OC}$↓   |
> | ---------- | :--------------: | :-------------: | :--------------: | :-------------: |
> | EOAPNet |   92.61(±3.13)   |   6.67(±2.91)   |  74.59(±25.64)   |   8.74(±5.34)   |
> | CBMT    |   93.36(±4.07)   |   6.20(±4.79)   |  81.16(±14.71)   |   8.37(±6.99)   |
> | **Ours**   | **94.17(±2.48)** | **5.15(±2.14)** | **82.74(±9.13)** | **7.52(±6.33)** |
>
> The results show that our method improves the performance of OD and OC segmentation by 0.81%(1.58%) Dice and 1.05(0.85) ASSD, respectively. Regarding the additional experiments on fundus image segmentation, we appreciate your interest and we will include these results in the supplementary materials of the revised paper.
>
> ---
>
> > **W4:** Feature visualizations
>
> As suggested, we provide some feature visualizations compared with the random masking strategy to more comprehensively demonstrate the effectiveness of the proposed complementary masking strategy. We provide the anonymous links for online viewing.
>
> For the direct comparison with random mask scheme, Figure R1 is: https://picx.zhimg.com/80/v2-c3ade1e071de958d79964bbb69c3564f.png. The input, feature and segmentation results are presented in dual-form. 'CMask' means using the proposed complementary masking strategy while 'RMask' means random masking.
>
> This is the results of the model trained with complementary masking strategy: https://picx.zhimg.com/80/v2-b6a712296ffc45377c54284a2ffb029b.png (Figure R2, last layer) and https://picx.zhimg.com/80/v2-603fcc5b6d192413c8ec988c28dffbc8.png (Figure R3, middle layer); this is the rusults of using random masking: https://picx.zhimg.com/80/v2-655ba0ffed4cbe7ad9168ad1e5f66e44.png (Figure R4, last layer) and https://picx.zhimg.com/80/v2-2957e2cca93f1cb93c05c95020c298e4.png (Figure R5, middle layer).
>
> ---
>
> Once again, thanks for your time and comments. We hope these concerns can be adequately addressed through the above explanations.
>
> ---
>
> **Reference**
>
> [1] Hoyer, Lukas, et al. "MIC: Masked image consistency for context-enhanced domain adaptation." Proceedings of the IEEE/CVF conference on computer vision and pattern recognition. 2023.

---

### Official Review · Reviewer_vim2 · 2025-03-12

**Overall Recommendation:** 3

**Summary:**

In this paper, the authors introduce MaskTwins, a UDA framework that integrates masked reconstruction into the main training pipeline. They argue that existing UDA methods leveraging masked image modeling treat masking merely as a form of input deformation and lack theoretical analysis, resulting in a superficial understanding and underutilization of its potential for feature extraction and representation learning. To address this, the authors reframe masked reconstruction as a sparse signal reconstruction problem and theoretically demonstrate that the dual form of complementary masks enhances the extraction of domain-agnostic image features. Experimental results on both natural and biological images show that MaskTwins improves domain generalization by enforcing consistency between predictions from complementary masked images, uncovering intrinsic structural patterns that persist across domains.

**Claims And Evidence:**

Most claims are valid. However, the authors state that they demonstrate the superiority of the proposed approach through extensive experiments. I disagree, as the experiments were conducted on a few relatively small datasets, and the domain gap between the source and target domains appears limited. This raises concerns about the generalizability of the proposed method.

**Essential References Not Discussed:**

Given the similarities between the proposed method and semi-supervised image segmentation, relevant works in semi-supervised image segmentation should be reviewed, and their connections and differences should be explicitly discussed.

**Experimental Designs Or Analyses:**

I reviewed the experimental design and found it generally reasonable. However, the small dataset sizes and limited source-target domain gap raise concerns about the generalizability of the results. Further evaluation on larger and more diverse datasets is needed to confirm the method’s effectiveness.

**Methods And Evaluation Criteria:**

Yes, the proposed methods and evaluation criteria are generally reasonable. However, their effectiveness remains to be evaluated on larger datasets with more significant source-target domain gaps to better assess their real-world applicability.

**Other Comments Or Suggestions:**

I have no other comments or suggestions.

**Other Strengths And Weaknesses:**

In addition to domain adaptation methods, I think it would be better if some semi-supervised learning methods can be used for comparison.

**Questions For Authors:**

The paper is clearly written. I think I only have the following questions:

1. Complementary masks provide tighter bounds, but how significant is the improvement? What is the ratio between the first and second terms in the bound shown in the SM for random masks, using typical values for the relevant symbols?

2. The results indicate that segmentation performance improves as the mask ratio increases from 0.1 to 0.5. Are these differences statistically significant?

3. Can the same conclusions hold across different architectures, such as CNNs, Transformers, and hybrid networks?

**Relation To Broader Scientific Literature:**

This work builds upon Hoyer et al.’s approach, which masks the target image and evaluates consistency between predictions from images with and without masking. It advances UDA by introducing a complementary masking strategy and providing a formal theoretical analysis to support the proposed method. The complementary masking strategy has been shown to improve domain generalization. Furthermore, the method closely aligns with semi-supervised image segmentation, differing mainly by incorporating an additional consistency loss that quantifies the discrepancy between predictions from masked target-domain images. While semi-supervised learning often assumes that labeled and unlabeled datasets originate from the same or similar domains, I argue that this assumption is not strictly necessary for semi-supervised methods to be effective. Therefore, the connection to semi-supervised image segmentation should be discussed more explicitly.

**Theoretical Claims:**

I briefly reviewed the proofs but did not conduct a thorough verification, so I cannot confirm their correctness.

---

> ### Author Rebuttal · Authors · 2025-04-01
>
> Thank you for your valuable time and comments. The main concerns are addressed below.
>
> > **W1:** Application on more challenging segmentation datasets
>
> Thank you for the concerns on the generalizability of the proposed method.
>
> On the one hand, the SYNTHIA dataset we use is of a relatively large scale, boasting over 20,000 synthetic images with diverse urban scene representations. As for the "SYNTHIA→Cityscapes" task, it has been extensively studied in domain adaptation [1], with a domain gap between synthetic and real-world data. On the other hand, biological image segmentation is commonly acknowledged to exhibit strong domain shifts between various domains.
>
> Still, we agree that extending our method to more challenging segmentation dataset can further validate its effectiveness. To address this, we conducted additional experiments on the RIM-ONE-r3 dataset for fundus optic disc (OD) and cup (OC) segmentation.
>
> ---
>
> Table R1. Quantitative results of OD and OC segmentation on the RIM-ONE-r3 dataset with a CBMT backbone, using only the proposed complementary masking strategy.
>
> |            |  $Dice_{OD}$↑  |   $ASSD_{OD}$↓    |  $Dice_{OC}$↑   |   $ASSD_{OC}$↓    |
> | ---------- | :--------------: | :-------------: | :--------------: | :-------------: |
> | EOAPNet |   92.61(±3.13)   |   6.67(±2.91)   |  74.59(±25.64)   |   8.74(±5.34)   |
> | CBMT[2] |   93.36(±4.07)   |   6.20(±4.79)   |  81.16(±14.71)   |   8.37(±6.99)   |
> | **Ours**   | **94.17(±2.48)** | **5.15(±2.14)** | **82.74(±9.13)** | **7.52(±6.33)** |
>
> The results show that our method improves the performance of OD and OC segmentation by 0.81%(1.58%) Dice and 1.05(0.85) ASSD, respectively.
>
> ---
>
> >**W2:** Connection with semi-supervised methods
>
> While we acknowledge the similarities between our method and semi-supervised image segmentation, they differ in task settings. Semi-supervised methods require labeled target data, whereas our focus is on unsupervised domain adaptation, especially under significant domain gaps. In the absence of labeled data, image-level masking plays a crucial role, and experiments focusing on this better validate our theoretical analysis.
>
> Still, we appreciate your interests and we will briefly discuss the connection with semi-supervised methods in the revised paper to strengthen the comprehensiveness and robustness of our research.
>
> ---
>
> >**Q1:** Gain of complementary masks and term ratios in formulas
>
> Intuitively, the generalization bound for random masking includes an additional term compared to complementary masking. While this extra term complicates the bound, complementary masking significantly tightens the generalization bound, leading to better performance across various scenarios.
>
> Directly quantifying this improvement from the formulas is complex due to multiple terms, but experimental results provide clarity. As shown in Table 1, MaskTwins outperforms MIC by +2.7 mIoU, demonstrating the effectiveness of our approach. Additionally, complementary masks contribute a +1.5 mIoU improvement over random masks (Table 4), confirming their value.
> Regarding the ratio of terms in the bounds, we observe that the additional term in random masking is influenced by the masking strategy. Complementary masking not only improves information preservation but also reduces generalization error variance, as stated in Theorem 4 of the supplementary material.
>
> Finally, "SM" does not appear in the paper, and we speculate the reviewer refers to the generalization bounds section, which discusses the benefits of complementary masking. The theory shows that complementary masking offers a tighter generalization bound compared to random masking.
>
> ---
>
> > **Q2:** Statistical results
>
> We provide the mean and standard deviation of the experiments as follows. The results show that the standard deviation of the metrics does not vary significantly with the change in the mask ratio.
>
>
> Table R2. Statistical results based on Table 5(a). "±" refers to the standard deviation over 3 random seeds.
>
> | Mask Ratio | mIoU       |
> | ---------- | ---------- |
> | 0.1        | 72.0(±0.2) |
> | 0.2        | 74.6(±0.3) |
> | 0.3        | 75.4(±0.2) |
> | 0.4        | 76.5(±0.2) |
> | 0.5        | 76.7(±0.2) |
>
> ---
>
> > **Q3:** Question on architectures
>
> The proposed complementary masking strategy certainly works on different architectures and we highlight that it does not rely on the network structure or additional internal components. Specifically, for natural and biological image segmentation, we use a Transformer network and a CNN network respectively, following the corresponding pipelines in [1]. The details of the architectures in the experiments can be found in Appendix F.
>
> **Reference**
>
> [1] Hoyer, Lukas, et al. "MIC: Masked image consistency for context-enhanced domain adaptation." CVPR 2023.
>
>
> [2] Tang L, Li K, He C, et al. Source-free domain adaptive fundus image segmentation with class-balanced mean teacher, MICCAI 2023.

---

> > ### Comment · Reviewer_vim2 · 2025-04-02
> >
> > 1. The authors did not directly address my question about statistical significance. Please use t-test or Wilcoxon signed rank test to assess significance.
> > 2. I disagree with the claim that semi-supervised methods require labeled target data. I would like to see results from a typical semi-supervised segmentation approach as a baseline for comparison.

---

> > > ### Author Response · Authors · 2025-04-05
> > >
> > > Thank you for your prompt feedback. We now fully and accurately understand your questions.
> > >
> > > >**1:** Statistical significance
> > >
> > > We agree that quantitatively evaluating by statistical significance can further demonstrate the performance improvement of complementary masking.
> > >
> > > Specifically, we use two related samples (complementary masks vs. random masks), each of size n=5. Initially, we verify whether the two sets of data are derived from a normal distribution by conducting the Shapiro-Wilk test. The corresponding p-values are 0.5767 and 0.5866. So we are inclined to accept the hypothesis of normal distribution at a confidence level of 0.05.
> > >
> > > Considering that the t-test is highly sensitive to the overall distribution characteristics of the data, we also perform the Wilcoxon signed-rank test in the one-sided case (the default two-sided case would result in a zero statistic value). Under both test scenarios, the p-value is lower than the significance level alpha=0.05. Hence, we draw the conclusion that the performance improvement of complementary masking is statistically significant.
> > >
> > > ---
> > > Table R3. The results of the t-test and Wilcoxon signed rank test (n=5).
> > >
> > > |    | statistic | p-value   |
> > > | - | - | -|
> > > | t-test   | 60.56 | 4.453e-07 |
> > > | Wilcoxon signed rank test | 15.0 | 0.03125 |
> > > ---
> > >
> > > > **2:** Semi-supervised methods for comparison
> > >
> > > We fully understand and agree with the reviewer's perspective after thorough research. To the best of our knowledge, only in the image classification task have we found that several articles [1, 2, 3] discuss and compare both unsupervised domain adaptation (UDA) and semi-supervised learning (SSL) methods.
> > >
> > > Both UDA and SSL aim for the model trained on labeled source data to perform well on target data. Intuitively, the difference is that SSL typically assumes the source domain is identical to the target domain. Additionally, UDA methods focus on minimizing the discrepancy between the two domains based on the theoretical bounds, while the motivation of SSL methods is the basic assumptions about the data structure.
> > >
> > > During the early developmental stage, UDA and SSL were developed independently for each specific setting. Subsequently, more and more researchers have realized that there is no technical barrier to applying SSL to UDA. In recent years, UDA methods have widely adopted SSL methods, such as mean teacher [4] and consistency regularization [5]. As a result, the boundary between UDA and SSL has become blurred.
> > >
> > > The existing explorations [1,2,3] have been predominantly focused on image classification tasks. Importantly, we have evaluated the effectiveness of our proposed method on the image classification task in the supplementary materials. Therefore, based on Table 7 in the supplementary materials, we have added a typical semi-supervised classification approach [1] as a baseline for comparison to address the reviewer's concerns.
> > >
> > > ---
> > >
> > > Table R4. The comparison results with a semi-supervised method A2LP [1] on VisDA-2017 with ResNet-101.
> > >
> > > |   | Acc. |
> > > | ---- | ---|
> > > | LP [1]   | 73.9  |
> > > | A2LP [1] | 82.7  |
> > > | **Ours** | **87.3**   |
> > > ---
> > >
> > > For image segmentation, we additionally provide the comparison with ILM-ASSL [6] which is one of the state-of-the-art semi-supervised methods on the leaderboard of the "Papers with Code" website. It follows a typical semi-supervised learning framework, combined with an uncertainty selection strategy of active learning.
> > >
> > > Compared with ILM-ASSL using 1% and 2.2% of labeled target data, our method consistently achieves higher performance, which strongly demonstrates the superiority of our proposed complementary masking strategy.
> > >
> > > ---
> > >
> > > Table R5. The comparison results with a semi-supervised method ILM-ASSL [6] on SYNTHIA-to-Cityscapes.
> > >
> > > |   | mIoU     |
> > > | ------ | ------ |
> > > | ILM-ASSL(1%)   | 73.2     |
> > > | ILM-ASSL(2.2%) | 76.0     |
> > > | **Ours**       | **76.7** |
> > > ---
> > >
> > > Once again, thanks for your time and comments. We hope these concerns can be adequately addressed through the above explanations.
> > >
> > > ---
> > >
> > > **Reference**
> > >
> > > [1] Zhang Y, Deng B, Jia K, et al. Label propagation with augmented anchors: A simple semi-supervised learning baseline for unsupervised domain adaptation. ECCV 2020.
> > >
> > > [2] Roelofs B, Berthelot D, Sohn K, et al. Adamatch: A unified approach to semi-supervised learning and domain adaptation. ICLR 2022.
> > >
> > > [3] Zhang Y, Zhang H, Deng B, et al. Semi-supervised models are strong unsupervised domain adaptation learners. arXiv preprint arXiv:2106.00417, 2021.
> > >
> > > [4] Tarvainen A, Valpola H. Mean teachers are better role models: Weight-averaged consistency targets improve semi-supervised deep learning results. NeurIPS 2017.
> > >
> > > [5] Samuli Laine and Timo Aila. Temporal ensembling for semi-supervised learning. ICLR 2017.
> > >
> > > [6] Guan L, Yuan X. Iterative loop method combining active and semi-supervised learning for domain adaptive semantic segmentation. arXiv preprint arXiv:2301.13361, 2023.

---

### Official Review · Reviewer_DCWw · 2025-03-13

**Overall Recommendation:** 3

**Summary:**

This paper introduces a complementary masking strategy for the semantic segmentation UDA task. Building on the existing masked image consistency (MIC) training paradigm, which relies on pseudo-labels from the teacher model, the authors propose a complementary masking loss to further enforce consistent predictions on complementary masked images. Additionally, the paper provides a theoretical analysis of the complementary masking strategy. Experiments demonstrate the effectiveness of the proposed approach.

**Claims And Evidence:**

Yes, the paper mainly proposes a complementary masking strategy for input images and provides a theoretical analysis of this approach.

**Essential References Not Discussed:**

I think this paper has included enough related works to provide a clear understanding

**Experimental Designs Or Analyses:**

In Table 4, although the ablation study is not directly discussed against the baseline MIC, Table 1 shows that simply replacing complementary masking with random masking (Row 3 in Table 4) already outperforms MIC (75.2 mIoU vs. 74.0 mIoU). This indicates that the performance gains in this paper heavily rely on the masking strategy itself, even when using random masking. Therefore, while the paper reports state-of-the-art results, these improvements are less convincing. Moreover, the authors do not provide an explanation for why masking strategies, including random ones, lead to such significant gains.

**Methods And Evaluation Criteria:**

This paper mainly introduces a complementary masking loss to encourage consistent predictions on complementary masked images. The model architecture follows the baseline MIC, consisting of a student and an EMA teacher. The masked consistency learning loss also follows the MIC loss from the baseline MIC (Hoyer et al., 2023), which uses pseudo-labels of the complete target image from the teacher for supervision. Notably, complementary masking strategies have already been explored in existing multi-modal works, both at the input and feature levels (e.g., Shin et al., 2024; Yang et al., 2025). Therefore, the primary contributions of this paper are its theoretical analysis and experimental validation, which demonstrate the effectiveness of the complementary masking strategy for RGB images alone. However, the method lacks sufficient novelty and insight.

**Other Comments Or Suggestions:**

No other comments

**Other Strengths And Weaknesses:**

Advantages:

1.The paper is well-written and clear.
2. The experiments are sufficient to evaluate the proposed method.
3. The paper provides detailed theoretical analysis and proofs.

Disadvantages:
1. As mentioned earlier, the method largely builds on strategies from existing baselines, with only minor modifications to achieve good performance, but lacks novel or insightful contributions.  I have another key question: how this masking strategy would perform in feature space, as both input and feature masking have been studied in multi-modal tasks. Similar concerns apply to the theoretical analysis.

**Questions For Authors:**

As mentioned in previous comments:
1. Explain why random masking alone can achieve state-of-the-art performance.
2. Discuss how the complementary masking strategy would perform in the feature space in this single-modal RGB setting.

**Relation To Broader Scientific Literature:**

1. Model architecture: Follows the baseline MIC, consisting of a student and an EMA teacher.
2. Masked consistency learning loss: Directly follows the MIC loss from the baseline (Hoyer et al., 2023), using pseudo-labels of the complete target image from the teacher for supervision.
3. Complementary masking strategy: Already explored in existing multi-modal works at both input and feature levels (e.g., Shin et al., 2024; Yang et al., 2025).

**Theoretical Claims:**

I have reviewed the theoretical claims.

---

> ### Author Rebuttal · Authors · 2025-04-01
>
> Thank you for your valuable time and comments. The main concerns are addressed below.
>
> > **W1:** Misunderstanding on Complementary Masking Techniques
>
> Thank you for your comments again. But we respectfully cannot agree with the justifications here, and would like to further clarify as below:
>
> - First, the proposed complementary masking is not limited to "RGB images alone"; it can also be applied to biological datasets that contain gray-scale images.
>
> - Meanwhile, we highlight our theoretical contributions, which are an indispensable part of our work and are entirely lacking in existing works. Specifically, we have provided a theoretical foundation for masked reconstruction from a perspective of sparse signal reconstruction problem and have rigorously analyzed the properties of complementary masking from three aspects: information preservation, generalization bound, and feature consistency.
>
> ---
>
> > **Q1:** Explanation on the gain of random masking over MIC
>
> The reason why random masking achieves a certain improvement compared to MIC (75.2 mIoU vs. 74.0 mIoU) is the addition of the dual-form masking consistency, while MIC only has a single masked image. Mask itself is indeed of great significance. However, the improvement brought about in this way is limited, and having more branches of masked images will also lead to the problem of redundancy. In our experiments, the performance of a single masked branch reached 74.0 mIoU, that of two branches improved to 75.2 mIoU and using three or more branches will not yield further gains.
>
> In contrast, the complementary masking can achieve significantly higher improvement by only changing the masking strategy (76.7 mIoU vs. 75.2 mIoU). This further validates the substantial performance boost brought about by the complementary masking and the total gain does not rely on the masking strategy itself.
>
> ---
>
> > **Q2:** The performance of complementary masking strategy in the feature space
>
> We are highly intrigued by the ideas you proposed, and as a result, we conducted the following experiments in Table R1. Since applying masking during the decoder phase may significantly impact the already-extracted high-level semantic information and lead to a decline in model performance, we perform masking on the four layers of encoded features (where 1 represents the shallowest layer and 4 represents the deepest layer) respectively.
>
>
> ---
> Table R1. The performance of complementary masking strategy in the feature space on SYNTHIA→Cityscapes.
>
> | Complementary Masking         | mIoU |
> | ----------------------------- | ---- |
> | image level                   | 76.7 |
> | feature level: 1 (shallowest) | 76.0 |
> | feature level: 2              | 74.6 |
> | feature level: 3              | 72.4 |
> | feature level: 4 (deepest)    | 67.7 |
> ---
>
> It is worth noting that the feature-level masking are mainly utilized in generative tasks, such as in MaskGit [1]. According to Assumption 1 in the main paper, latent-level masking may result in more information loss. Therefore, it is not suitable for comprehension tasks, for example, image segmentation.
>
> The above results also confirm this. The deepest feature layer contains the most semantic information and applying complementary masking to this layer will result in the most substantial loss of valuable details, leading to the most significant drop in performance.
>
> Regarding the additional experiments on feature-level masking, we appreciate your interest and will include these results in the supplementary materials of the revised paper.
>
> ---
>
> **Reference**
>
> [1] Chang H, Zhang H, Jiang L, et al. Maskgit: Masked generative image transformer[C]//Proceedings of the IEEE/CVF conference on computer vision and pattern recognition. 2022: 11315-11325.

---

> > ### Comment · Reviewer_DCWw · 2025-04-05
> >
> > Thank you to the authors for the rebuttals to my reviews. After reviewing the responses, I think most of my concerns have been addressed. Therefore, I am changing my score from  "2: Weak reject " to "3: Weak accept".
> >
> > I would like to clarify a point regarding my comment in W1. I understand that the authors have conducted experiments using grayscale images. In my previous review, I mentioned that "this paper demonstrates the effectiveness of the complementary masking strategy for RGB images alone." By "RGB images alone," I meant that the complementary masking strategy in this paper is applied solely in the input image space, rather than in the feature space. As noted in my review, "complementary masking strategies have already been explored in existing multi-modal works, both at the input and feature levels." I also raised this as a formal question in Q2 to encourage further discussion, and I appreciate the authors’ inclusion of results and discussions on feature-level masking.

---

> > > ### Author Response · Authors · 2025-04-06
> > >
> > > Thank you for your clarification and updated score! We're glad our response addressed your concerns and that we had the opportunity to discuss feature-level masking. Once again, we sincerely appreciate your time and effort in reviewing our paper and reading our comments!

---

### Official Review · Reviewer_sqBy · 2025-03-14

**Overall Recommendation:** 3

**Summary:**

This paper explores the connection between Masked Image Modeling (MIM) and consistency regularization in Unsupervised Domain Adaptation (UDA). It reframes masked reconstruction as a sparse signal reconstruction problem and theoretically proves that complementary masks can effectively extract domain-agnostic features. Based on this insight, the authors propose MaskTwins, a UDA framework that integrates masked reconstruction into the main training pipeline. MaskTwins enforces consistency between predictions of images masked in complementary ways, uncovering intrinsic structural patterns across domains and enabling end-to-end domain generalization. Extensive experiments on natural and biological image segmentation demonstrate its superiority over baseline methods, highlighting its effectiveness in extracting domain-invariant features without separate pre-training.

**Claims And Evidence:**

Yes.

**Essential References Not Discussed:**

No.

**Experimental Designs Or Analyses:**

Yes.

**Methods And Evaluation Criteria:**

Yes.

**Other Comments Or Suggestions:**

Some of the statements in this paper need refinement. For example, the first contribution in the introduction appears to be an overclaim, and the subsequent analysis lacks sufficient detail.

**Other Strengths And Weaknesses:**

Strengths:
The paper is well-organized, with a clear and intuitive research motivation. It provides comprehensive theoretical analysis, and the proposed MaskTwins framework is validated through extensive experimental results.

Weaknesses:
1. Although the paper offers thorough theoretical analysis, the proposed MaskTwins framework lacks architectural innovation. The ideas related to Masked Image Modeling (MIM) are not novel.
2. The ablation study in Table 4 is not comprehensive. Ablation experiments should control for single variables, but the current design lacks specificity.
3. The formatting of Table 4 and Table 5 is inconsistent with other tables in the paper, such as the use of bold text with a gray background.

**Questions For Authors:**

N/A

**Relation To Broader Scientific Literature:**

The key contributions of this paper relate to the broader scientific literature by advancing the understanding of masked image modeling (MIM) in the context of unsupervised domain adaptation (UDA). It builds on prior works that connect MIM and consistency regularization but goes further by reframing masked reconstruction as a sparse signal problem and proving the effectiveness of complementary masks for domain-agnostic feature extraction. This theoretical grounding differentiates it from previous methods that treat masking superficially. The proposed MaskTwins framework integrates these insights directly into the training pipeline, offering a novel approach to domain generalization.

**Theoretical Claims:**

Yes.

---

> ### Author Rebuttal · Authors · 2025-04-01
>
> Thank you for your valuable time and comments. The main concerns are addressed below.
> >**W1:** Misunderstanding on Complementary Masking Techniques
>
> First, the ideas related to Masked Image Modeling (MIM) can be novel. For instance, the latest ECCV 2024 paper MICDrop [1] explored the masking of multi-modal information, indicating continuous innovation within the MIM field. And MIM serves as the foundation of our approach. Based on this, our work focus on the extension to domain adaptative tasks and the exploration of the upper bounds of performance using the proposed complementary masking strategy. Meanwhile, the improvements in our approach stem from both the dual-form and the framework. Moreover, we conducted a series of additional experiments specifically on the masks themselves to validate our theoretical analysis.
>
> We highlight our theoretical contributions, which are an indispensable part of our work and are entirely lacking in existing works. Specifically, we have provided a theoretical foundation for masked reconstruction from a perspective of sparse signal reconstruction problem and have rigorously analyzed the properties of complementary masking from three aspects: information preservation, generalization bound, and feature consistency. For more theoretical details, please see the supplementary materials.
>
> ---
>
> > **W2:** More ablation experiments
>
> In Table 4, we focus on exploring the individual effects of each component by controlling for single variables. More importantly, we aim to compare with random masking to validate the effectiveness of the proposed complementary masking strategy and support our theoretical analysis. Still, to conduct more rigorous experiments, we additionally ablated complementary mask-only and the effect of EMA and AdaIN using random masking strategy.
>
> ---
>
> Table R1. Supplementary ablation rxperiment results for Table 4. The last four rows are newly added experimental data.
>
> |  CL  | CMask | RMask | EMA  | AdaIN | mIoU |
> | :--: | :---: | :---: | :--: | :---: | :--: |
> |  -   |   -   |   -   |  -   |   -   | 53.7 |
> |  √   |   -   |   -   |  √   |   √   | 72.8 |
> |  √   |   -   |   √   |  √   |   √   | 75.2 |
> |  √   |   √   |   -   |  √   |   √   | 76.7 |
> |  √   |   √   |   -   |  -   |   √   | 76.1 |
> |  √   |   √   |   -   |  √   |   -   | 76.4 |
> |  √   |   √   |   -   |  -   |   -   | 76.0 |
> |  √   |   -   |   √   |  √   |   -   | 75.0 |
> |  √   |   -   |   √   |  -   |   √   | 74.6 |
> |  √   |   -   |   √   |  -   |   -   | 74.3 |
> ---
> > **W3:** Revision of the formatting of tables
>
> Thank you for pointing it out. We will reorganize the formatting of tables carefully in the revised paper.
>
> ---
>
> **Reference**
>
> [1] Yang L, Hoyer L, Weber M, et al. Micdrop: masking image and depth features via complementary dropout for domain-adaptive semantic segmentation[C]//European Conference on Computer Vision. Cham: Springer Nature Switzerland, 2024: 329-346.

---

### Decision · Program_Chairs · 2025-05-01

**Decision:**

Accept (poster)

**Comment:**

Dear authors,

Thank you for submitting draft for review. This draft has received 4 weak accept. One reviewer improved from weak reject to weak accept after rebuttal. However, none of the reviewer is interested in assigning accept rating.  One reviewer pointed out concern regarding whether it can handle considerable domain shift. Method itself is quite straightforward,  however,  notion that just use of complimentary masks can result in this considerable improvement is quiet interesting. We will encourage authors to include all the points discussed during rebuttal in the final copy, especially the updated ablation studies.



regards
AC